# Conformal mapping Coordinates Physics-Informed Neural Networks (CoCo-PINNs): learning neural networks for designing neutral inclusions

## Abstract

We focus on designing and solving the neutral inclusion problem via neural networks. The neutral inclusion problem has a long history in the theory of composite materials, and it is exceedingly challenging to identify the precise condition that precipitates a general-shaped inclusion into a neutral inclusion. Physics-informed neural networks (PINNs) have recently become a highly successful approach to addressing both forward and inverse problems associated with partial differential equations. We found that traditional PINNs perform inadequately when applied to the inverse problem of designing neutral inclusions with arbitrary shapes. In this study, we introduce a novel approach, Conformal mapping Coordinates Physics-Informed Neural Networks (CoCo-PINNs), which integrates complex analysis techniques into PINNs. This method exhibits strong performance in solving forward-inverse problems to construct neutral inclusions of arbitrary shapes in two dimensions, where the imperfect interface condition on the inclusion's boundary is modeled by training neural networks. Notably, we mathematically prove that training with a single linear field is sufficient to achieve neutrality for untrained linear fields in arbitrary directions, given a minor assumption. We demonstrate that CoCo-PINNs offer enhanced performances in terms of credibility, consistency, and stability.

## 1 Introduction

Physics-informed neural networks (PINNs)(Raissi et al., 2019; Karniadakis et al., 2021) are specialized neural networks designed to solve partial differential equations (PDEs). Since their introduction, PINNs have been successfully applied to a wide range of PDE-related problems (Cuomo et al., 2022; Hao et al., 2023; Wu et al., 2024). A significant advantage of using PINNs is their versatile applicability to different types of PDEs and their ability to deal with PDE parameters or initial/boundary constraints while solving forward problems (Akhound-Sadegh et al., 2023; Cho et al., 2023; Rathore et al., 2024; Lau et al., 2024). The conventional approach to solving inverse problems with PINNs involves designing neural networks that converge to the parameters or constraints to be reconstructed, which are typically modeled as constants or functions. We refer to this methodology as *classical PINNs*. Numerous successful outcomes in solving inverse problems using PINNs have been reported. See, for example, Chen et al. (2020); Jagtap et al. (2022); Haghighat et al. (2021). However, as the complexity of the PDE-based inverse problem increases, the neural networks may require additional design to represent the parameters or constraints accurately. For instance, Pokkunuru et al. (2023) utilized Bayesian approach to design the loss function, Guo et al. (2022) used Monte Carlo approximation to compute the fractional derivatives, Xu et al. (2023) adopted multi-task learning method to weight losses and also presented the forward-inverse problem combined neural networks, and Yuan et al. (2022) propose the auxiliary-PINNs to solve the forward and inverse problems of integro-differential equations. This increase in network complexity can significantly escalate computational difficulties and the volume of data necessary for training PINNs. Moreover, the direct approach to approximate the reconstructing parameters by neural networks enables too much fluent representation ability. This alludes to the fact that the conventional approach is inadequate depending on the problems due to the intrinsic ill-posedness of inverse problems.

In this paper, we apply the PINNs' framework to address the inverse problem of designing neutral inclusions, a topic that will be elaborated below. The challenge of designing neutral inclusions falls within the scope where traditional PINNs tend to perform inadequately. To overcome this limitation, we propose improvements to the PINNs approach by incorporating mathematical analytical methods.

Inclusions with different material features from the background medium commonly cause perturbations in applied fields when they are inserted into the medium. Problems analyzing and manipulating the effects of inclusions have gained significant attention due to their fundamental importance in the modeling of composite materials, particularly in light of rapid advancements and diverse applications of these materials. Specific inclusions, referred to as *neutral inclusions*, do not disturb linear fields (see Figure 2). The neutral inclusion problem has a long and established history in the theory of composite materials (Milton, 2002). Some of the most well-known examples

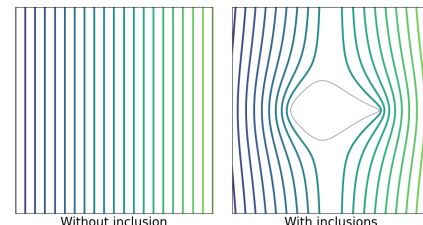

Figure 1: Inclusion makes perturbation.

include coated disks and spheres (Hashin, 1962; Hashin & Shtrikman, 1962), as well as coated ellipses and ellipsoids (Grabovsky & Kohn, 1995; Kerker, 1975; Sihvola, 1997). The primary motivation for studying neutral inclusions is to design reinforced or embedded composite materials in such a way that the stress field remains unchanged from that of the material without inclusions and avoids stress concentration. Extensive research has been conducted on neutral inclusions and related concepts, such as invisibility cloaking involving wave propagation, in fields including acoustics, elasticity, electromagnetic waves within the microwave range Alù & Engheta (2005); Ammari et al. (2013); Landy & Smith (2013); Liu et al. (2017); Zhou & Hu (2006; 2007); Zhou et al. (2008); Yuste et al. (2018).

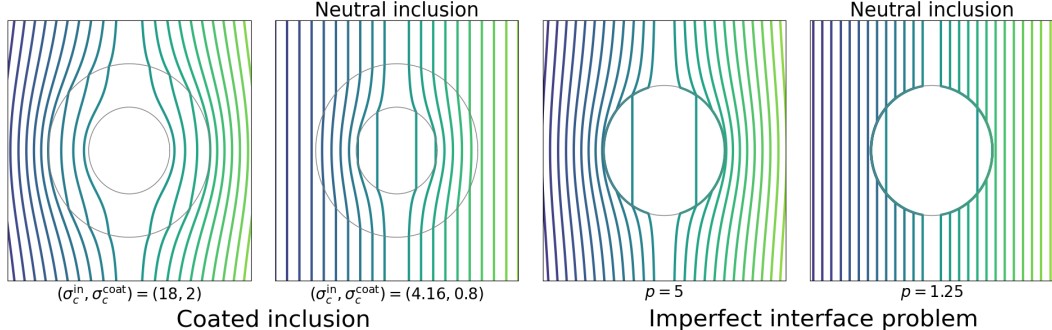

Figure 2: Neutral inclusions with circular shapes.

Designing neutral inclusions with general shapes presents an inherent challenge. In the context of the conductivity problem, which is the focus of this paper, mathematical theory shows that only coated ellipses and ellipsoids can maintain neutral properties for linear fields in all directions (Kang & Lee, 2014; Kang et al., 2016; Milton & Serkov, 2001). In contrast, non-elliptical shapes can remain neutral for just a single linear field direction (Jarczyk & Mityushev, 2012; Milton & Serkov, 2001). To address the difficulty of designing general shaped neutral inclusions, relaxed versions of the problem have been studied (Choi et al., 2023; Kang et al., 2022; Lim & Milton, 2020). Differently from the above examples, where perfectly bonding boundaries are assumed, imperfect interfaces introduce discontinuities in either the flux or potential boundary conditions in PDEs. Ru (1998) found interface parameters for typical inclusion shapes in two-dimensional elasticity for typical inclusion shapes. Benveniste & Miloh (1999) found neutral inclusions under a single linear field. The interface parameters, which characterize these discontinuities, may be non-constant functions defined along the boundaries of the inclusions so that, theoretically, the degree of freedom of the interface parameters is infinite. Hence, we expect to overcome the inherent challenge of designing neutral inclusions with general shapes by considering neutral inclusions with imperfect interfaces. A powerful technique for dealing with planar inclusion problems of general shapes has been to use conformal mappings and to define orthogonal curvilinear coordinates (Movchan & Serkov, 1997; Cherkaev

et al., 2022; Ammari & Kang, 2004; Jung & Lim, 2021), where the existence of the conformal mapping for a simply connected bounded domain is mathematically guaranteed by the Riemann mapping theory. Using these coordinates, Kang & Li (2019); Choi & Lim (2024) constructed weakly neutral inclusions that yield zero coefficients for leading-order terms in PDEs solution expansions (also refer to Milton (2002); Choi et al. (2023); Lim & Milton (2020) for neutral inclusion problems using the conformal mapping technique). However, such asymptotic approaches cannot achieve complete neutrality within this framework. Moreover, the requirement for analytic asymptotic expressions poses limitations on the generalizability of this approach.

In this paper, by adopting a deep learning approach approach, we focus on precise values of the solution rather than asymptotic ones. Unlike the asymptotic approaches in Kang & Li (2019); Choi & Lim (2024), the proposed method does not rely on an analytical expansion formula and incorporates the actual solution values directly into the loss function design. More precisely, we introduce a novel forward-inverse PINN framework by combining complex analysis techniques into PINNs, namely Conformal mapping Coordinates Physics-Informed Neural Networks (CoCo-PINNs). We define the loss function to include the evaluations of the solutions at sample points exterior of inclusions. Furthermore, we leverage the conformal mapping to effectively sample collocation points for PDEs involving general shaped inclusions. We found that classical PINNs–which treat the interface parameters as functions approximated by neural networks–perform inadequately when applied to designing imperfect parameters for achieving neutrality. Instead, we propose training the Fourier series coefficients of the imperfect parameters, rather than approximating the function. We test the performance of our proposed method in finding the forward solution by using the analytical mathematical results for the forward solution presented in Choi & Lim (2024), where theoretical direct solutions are expressed as products of infinite dimensional matrices whose entries depends on the expansion coefficients of interface parameter. Additionally, we leverage these analytical results to explain that why it is possible to train the PINN for the neutral inclusion using only a single applied field (see Subsection 3.2).

Many PINNs approaches to solving PDEs focus primarily on the forward problem and are typically validated through comparisons with numerical methods such as Finite Element Methods (FEM) and Boundary Element Methods (BEM), and others. In contrast, our problem addresses both forward and inverse problems simultaneously, adding complexity, especially in cases involving complex-shaped inclusions, and making it more challenging to achieve accurate forward solutions. Consequently, "reliability" becomes a critical factor when applying PINNs to our problem. The proposed CoCo-PINNs provide more accurate forward solutions, along with improved identification of the inverse parameters, compared to classical PINNs. They demonstrate greater consistency in repeated experiments and exhibit improved stability with respect to different conductivity values $\sigma_c$. We conduct experiments to ensure the "reliability" of CoCo-PINNs by assessing the credibility, consistency, and stability.

It is noteworthy that by utilizing Fourier series expansions to represent the inverse parameters, CoCo-PINNs offer deeper analytic insights into these parameters, making the solutions not only more accurate but also more explainable. Furthermore, our method requires no training data for the neutral inclusion problem due to its unique structure, where constraints at exterior points effectively serve as data. An additional remarkable feature is that our proposed method has proven effective in identifying optimal inverse parameters that are valid for general first-order background fields including those not previously trained. This impressive result is supported by a rigorous mathematical analysis.

In summary, this paper contains the following contributions:

- We developed a novel approach to PINNs, namely CoCo-PINNs, which have been shown to offer enhanced credibility, consistency, and stability compared to classical PINNs.

- We have adopted the exterior conformal mapping in the PINNs to make it train the problem corresponding to the arbitrarily shaped domains.

- Due to the nature of the neutral inclusion problem, it can be mathematically shown that once training with a given background solution $H(x) = x_1$ or $x_2$, a similar effect can be obtained for arbitrary linear background solutions $H(x) = ax_1 + bx_2$ for any $a, b \in \mathbb{R}$ (see Section 3.2 for more details).

## 2 INVERSE PROBLEM OF NEUTRAL INCLUSIONS WITH IMPERFECT CONDITIONS

We set $x = (x_1, x_2)$ to be a two-dimensional vector in $\mathbb{R}^2$. On occasion, we regard $\mathbb{R}^2 \cong \mathbb{C}$, whereby $x = x_1 + ix_2$ will be used. We assume that $\Omega \subsetneq \mathbb{C}$ is a nonempty simply connected bounded domain with an analytic boundary (refer Appendix B). We assume the interior region $\Omega$ has the constant conductivity $\sigma_c$ while the background medium $\mathbb{R}^2 \setminus \overline{\Omega}$ has the constant conductivity $\sigma_m$. We set $\sigma = \sigma_c \chi_\Omega + \sigma_m \chi_{\mathbb{R}^2 \setminus \overline{\Omega}}$, where $\chi$ is the characteristic function. We further assume that the boundary of $\partial\Omega$ is not perfectly bonding, resulting in a discontinuity in the potential. This discontinuity is represented by a nonnegative real-valued function $p$ on $\partial\Omega$, referred to as the *interface parameter or interface function*. Specifically, we consider the following potential problem:

$$\begin{cases} \nabla \cdot \sigma \nabla u = 0 & \text{in } \mathbb{R}^2, \\ p(u|^+ - u|^-) = \sigma_m \frac{\partial u}{\partial \nu}\big|^+ & \text{on } \partial\Omega, \\ \sigma_m \frac{\partial u}{\partial \nu}\big|^+ = \sigma_c \frac{\partial u}{\partial \nu}\big|^- & \text{on } \partial\Omega, \\ (u - H)(x) = O(|x|^{-1}) & \text{as } |x| \to \infty, \end{cases} \tag{1}$$

where $H$ is an applied background potential and $\partial u / \partial \nu$ denotes the normal derivative $\partial u / \partial \nu = \langle \nabla u, N \rangle$ with the unit exterior normal vector $N$ to $\partial\Omega$.

Here, we define neutral inclusion to provide a clearer explanation.

**Definition 1.** *We define $\Omega$ as a neutral inclusion for the imperfect interface problem equation 1 if $(u - H)(x) = 0$ for all $x$ in the exterior region $\mathbb{R}^2 \setminus \overline{\Omega}$ where $H(x)$ is any arbitrary linear fields, i.e., $H(x) = ax_1 + bx_2$ for any $a, b \in \mathbb{R}$.*

We explore the development of neural networks to find a specific interface function $p$ that makes $\Omega$ a neutral inclusion, given the inclusion $\Omega$ along with the conductivities $\sigma_c$ and $\sigma_m$, while simultaneously providing the forward solution $u$.

### 2.1 SERIES SOLUTION FOR THE GOVERNING EQUATION VIA CONFORMAL MAPPING

By the Riemann mapping theorem (see Appendix B), there exists a unique $\gamma > 0$ and conformal mapping $\Psi$ from $D = \{w \in \mathbb{C} : |w| > \gamma\}$ onto $\mathbb{C} \setminus \overline{\Omega}$ such that $\Psi(\infty) = \infty$, $\Psi'(\infty) = 1$, and

$$\Psi(w) = w + a_0 + \frac{a_1}{w} + \frac{a_2}{w^2} + \cdots . \tag{2}$$

We set $\rho_0 = \ln \gamma$. We use modified polar coordinates $(\rho, \theta) \in [\rho_0, \infty) \times [0, 2\pi)$ via $z = \Psi(w) = \Psi(e^{\rho + i\theta})$. One can numerically compute the conformal mapping coefficients $\gamma$ and $a_n$ for a given domain $\Omega$ (Jung & Lim, 2021; Wala & Klöckner, 2018). In the following, we assume $\Psi$ is given. Choi & Lim (2024) obtained the solution expression for the solution $u$ using the geometric function theory for complex analytic functions for a given arbitrary analytic domain $\Omega$:

**Theorem 1** (Analytic solution formula). *Let $F_m(z)$ be the Faber polynomials associated with $\Omega$ and the applied field is given by $H(z) = \Re\left(\sum_{m=1}^{\infty} \alpha_m F_m(z)\right)$, the solution $u$ satisfies*

$$(u - H)(z) = \Re\left(\sum_{m=1}^{\infty} \sum_{n=1}^{\infty} s_{mn} w^{-n}\right) \quad \text{for } |w| > \gamma, \tag{3}$$

$$\boldsymbol{s} := \{s_{mn}\}_{m,n \geq 1} = -\boldsymbol{\alpha}\widetilde{A}_1 - \overline{\boldsymbol{\alpha}}\widetilde{A}_2, \tag{4}$$

*where $\boldsymbol{\alpha}$ is a diagonal matrix whose entries are $\alpha_m$, and $\Re(\cdot)$ denotes the real part of a complex number. The matrices $\widetilde{A}_1$ and $\widetilde{A}_2$ are determined by the conductivities, the shape of inclusion, and the coefficients of the expansion formula of the interface function $p(z)$; refer to Appendix B for the concept of Faber polynomials and explicit definitions.*

From Theorem 1, the exact solution $u$ to Eq. (1) can be obtained analytically. We denote this solution as $u_p$ for comparison with the trained forward solution by neural networks. This analytic solution $u_p$ will be used to evaluate the credibility of the PINNs forward solver, as discussed in Section 4.2.

## 2.2 SERIES REPRESENTATION OF THE INTERFACE FUNCTION

In this subsection, based on complex analysis, we present the series expansion formula for designing the CoCo-PINNs. We assume that the interface function $p(x)$ is nonnegative, bounded, and continuous on $\partial\Omega$. By introducing a parametrization $x(\theta)$ of $\partial\Omega$ with $\theta \in [0, 2\pi)$, the interface function admits a Fourier series expansion with respect to $\theta$:

$$p(x(\theta)) = a_0 + \sum_{k \in \mathbb{N}} \left(a_k \cos(k\theta) + b_k \sin(k\theta)\right), \quad \theta \in [0, 2\pi),$$

where $a_k$ and $b_k$ are real constant coefficients. In particular, we take $x(\theta) = \Psi(w)$, $w = \gamma e^{i\theta}$, where $\Psi$ is the conformal mapping in Eq. (2). In this case, we have

$$\begin{aligned} p(w) := p(x(\theta)) &= \Re\left(p_0 + p_1 w + p_2 w^2 + p_3 w^3 + \cdots\right) \\ &= p_0 + p_1 w + \overline{p_1}\gamma^2 w^{-1} + p_2 w^2 + \overline{p_2}\gamma^4 w^{-2} + \cdots, \quad |w| = \gamma, \end{aligned} \tag{5}$$

for some complex-valued constants $p_k$. Note that we can similarly express the boundary conditions in Eq. (1) in terms of the variable $w$, enabling us to effectively address these boundary conditions.

We further assume that $p(w)$ is represented by a finite series, truncated at the $w^n$-term for some $n \in \mathbb{N}$. Specifically, we define

$$p^{(n)}(w) = \Re\left(\sum_{k=0}^{n} p_k w^k\right). \tag{6}$$

At this state, the reconstruction parameters are deduced to $p_0, \cdots, p_n$, and this makes the inverse problem of determining the interface parameter over-determined by using the constraints $(u - H)(x) \approx 0$ for many sample points exterior of $\Omega$.

A fundamental characteristic of inverse problems is that they are inherently ill-posed, and the existence or uniqueness of the inverse solution–the interface function in this paper–is generally not guaranteed. When solving a minimization problem, neural networks may struggle if the problem admits multiple minimizers. In particular, the ability of neural networks to approximate a wide range of functions can lead them to converge to suboptimal solutions, corresponding to local minimizers of the loss function. As a result, the classical approach in PINNs, which allows for flexible function representation, can be highly sensitive to the initial parameterization. In contrast, the series expansion approach constrains the solution to well-behaved functions, reducing sensitivity to the initial parameterization and ensuring the regularity of the target function. Additionally, since the series approximation method requires fewer parameters, it can be treated as an over-determined problem, helping to mitigate the ill-posedness of inverse problems.

## 3 THE PROPOSED METHOD: CoCo-PINNS

This section introduces the CoCo-PINNs, their advantages, and the mathematical reasoning behind why neutral inclusions designed by training remain effective even in untrained background fields. We begin with the loss design corresponding to the imperfect interface problem Eq. (1), whose solution exhibits a discontinuity across $\partial\Omega$ due to the imposed boundary conditions. We denote the solutions inside and outside $\Omega$ as $u^{\text{int}}$ and $u^{\text{ext}}$, respectively, and represent their neural networks' approximations as $u_{\text{NN}}^{\text{int}}$ and $u_{\text{NN}}^{\text{ext}}$. We named these solutions as trained forward solutions. We aim to train the interface function, represented by $p^{(n)}$ for the truncated series approximation and $p_{\text{NN}}$ for the fully connected neural network approximation. The method utilizing $p^{(n)}$ is referred to as CoCo-PINNs, while the approach using $p_{\text{NN}}$ to represent the interface function is called classical PINNs.

### 3.1 MODEL DESIGN FOR THE FORWARD-INVERSE PROBLEM

We utilize three sets of collocation points: $\Omega^{\text{int}}$, $\Omega^{\text{ext}}$, and $\partial\Omega$, which are finite sets of points corresponding to the interior, exterior, and boundary of $\Omega$, respectively, with a slight abuse of notation for $\partial\Omega$. We select collocation points based on conformal mapping theory to handle PINNs in arbitrarily shaped domains, and provide a detailed methodology for this selection in Appendix D.1. To address the boundary conditions for $z \in \partial\Omega$, we use $x_z = z + \delta N$ and $y_z = z - \delta N$ for the limit of the boundary from the exterior and interior, respectively, with small $\delta > 0$, and unit normal vector $N$.

The loss functions corresponding to the governing equation and the design of a neutral inclusion are defined as follows:

$$\mathcal{L}_{\text{PDE}}^{\text{int}} = \nabla \cdot \sigma_c \nabla u_{\text{NN}}^{\text{int}} \qquad\qquad \text{for } x \in \Omega^{\text{int}}, \tag{7}$$

$$\mathcal{L}_{\text{PDE}}^{\text{ext}} = \nabla \cdot \sigma_m \nabla u_{\text{NN}}^{\text{ext}} \qquad\qquad \text{for } x \in \Omega^{\text{ext}}, \tag{8}$$

$$\mathcal{L}_{\text{bd}}^{(1)} = p^{(n)} \left( u_{\text{NN}}^{\text{ext}} - u_{\text{NN}}^{\text{int}} \right) - \sigma_m \frac{\partial u_{\text{NN}}^{\text{ext}}}{\partial \nu}(x_z) \qquad\qquad \text{for } z \in \partial\Omega, \tag{9}$$

$$\mathcal{L}_{\text{bd}}^{(2)} = \sigma_m \frac{\partial u_{\text{NN}}^{\text{ext}}}{\partial \nu}(x_z) - \sigma_c \frac{\partial u_{\text{NN}}^{\text{int}}}{\partial \nu}(y_z) \qquad\qquad \text{for } z \in \partial\Omega, \tag{10}$$

$$\mathcal{L}_{\text{Neutral}} = u_{\text{NN}}^{\text{ext}} - H \qquad\qquad \text{for } x \in \Omega^{\text{ext}}. \tag{11}$$

with $\partial u/\partial \nu = \langle \nabla u, N \rangle$. In the case where we train using classical PINNs, we replace the interface function $p^{(n)}$ with $p_{\text{NN}}$. To enforce the non-negativity of the interface function, we introduce an additional loss function $\mathcal{L}_{\text{plus}} = \max\{0, -p\}$.

By combining all the loss functions with weight variables $\{w_i\}_{i=1}^5$, we define the total loss by

$$\mathcal{L}_{\text{Total}} = \frac{w_1}{|\Omega^{\text{int}}|} \sum_{x \in \Omega^{\text{int}}} \left( \mathcal{L}_{\text{PDE}}^{\text{int}} \right)^2 + \frac{w_1}{|\Omega^{\text{ext}}|} \sum_{x \in \Omega^{\text{ext}}} \left( \mathcal{L}_{\text{PDE}}^{\text{ext}} \right)^2 + \frac{w_2}{|\partial\Omega|} \sum_{z \in \partial\Omega} \left( \mathcal{L}_{\text{bd}}^{(1)} \right)^2 $$
$$+ \frac{w_3}{|\partial\Omega|} \sum_{z \in \partial\Omega} \left( \mathcal{L}_{\text{bd}}^{(2)} \right)^2 + \frac{w_4}{|\Omega^{\text{ext}}|} \sum_{x \in \Omega^{\text{ext}}} \left( \mathcal{L}_{\text{Neutral}} \right)^2 + w_5 \sum_{x \in \partial\Omega} \mathcal{L}_{\text{plus}}. \tag{12}$$

Here, $|A|$ denotes the number of the elements in the set $A$. We then consider the following loss with regularization term:

$$\mathcal{L}_{\text{Reg}} = \begin{cases} \mathcal{L}_{\text{total}} + \epsilon \left( 2\pi\gamma^2 |p_0|^2 + 4\pi\gamma^2 \sum_{k=1}^n (1+k^2)|p_k|^2 \right), & p = p^{(n)}, \\ \mathcal{L}_{\text{total}} + \epsilon \|\mathbf{w}_p\|_F^2, & p = p_{\text{NN}}, \end{cases} \tag{13}$$

where $\mathbf{w}_p$ represents the weights of the neural networks $p_{\text{NN}}$, $\| \cdot \|_F$ is the Frobenius norm, and the $W^{1,2}(\partial\Omega)$-norm is used for $p^{(n)}$, that is,

$$\|p^{(n)}\|_{W^{1,2}(\partial\Omega)}^2 = \|p^{(n)}\|_{L^2(\partial\Omega)}^2 + \|\nabla p^{(n)}\|_{L^2(\partial\Omega)}^2 = 2\pi\gamma^2 |p_0|^2 + 4\pi\gamma^2 \sum_{k=1}^n (1+k^2)|p_k|^2.$$

This type of regularization is commonly used to address ill-posed problems. We used the loss in Eq. (13) for all experiments.

CoCo-PINNs are designed using complex geometric function theory to address the interface problem. While classical PINNs rely on neural network approximations based on the universal approximation theory, CoCo-PINNs utilize Fourier series expansion, which helps overcome the challenges of ill-posedness in the neutral inclusion inverse problems and ensures that the inverse solution remains smooth. Additionally, this approach allows for the selection of initial coefficients of the interface function using mathematical results. The results of CoCo-PINNs can be explained by a solid mathematical foundation, as discussed in Section 3.2. In Section 4, we examine the advantages of CoCo-PINNs in terms of credibility, consistency, and stability.

### 3.2 NEUTRAL INCLUSION EFFECTS FOR UNTRAINED LINEAR FIELDS

In this section, we briefly explain the reason that the CoCo-PINNs can yield the neutral inclusion effect for untrained background fields. Since the governing equation Eq. (1) is linear with respect to $H$, the following trivially holds by the properties of linear PDEs:

**Theorem 2.** *Consider a domain, denoted by $\Omega$, that is of arbitrary shape and whose boundary is given by an exterior conformal mapping $\Psi(w)$. If there exists an interface function $p(x)$ that makes $\Omega$ a neutral inclusion for the background field $H(x) = x_1$ and $x_2$, simultaneously, then $\Omega$ is neutral also for all linear fields $H(x) = ax_1 + bx_2$ of arbitrary directions $(a, b) \in \mathbb{R}^2$.*

By Theorem 2, one can expect to find a function $p(x)$ such that $\Omega$ is neutral to all linear fields $H$ by training with only two background fields, assuming such a $p(x)$ exists. Although the existence of this function has not yet been theoretically verified, experiments in this paper with various shapes demonstrate that, for given $\Omega$, there exists a $p(x)$ that produces the *neutral inclusion effect*, meaning that the perturbation $u - H$ is negligible for all directions of $H$.

**Remark 1.** *In all examples, we successfully identified a $p(x)$ with the neutral inclusion effect by training with only a single $H$. These results can be explained by the following theorem.*

**Theorem 3.** *Let $\Omega$ with an interface function $p(x)$ be neutral for a single background field $H$. If the first rows of $\widetilde{A}_1$ and $\widetilde{A}_2$ given in Eq. (4) are linearly independent, then $\Omega$ is neutral also for all linear fields $H(x) = ax_1 + bx_2$ of arbitrary directions $(a,b) \in \mathbb{R}^2$.*

The proof of Theorem 3 can be found in Appendix C. As a future direction, it would be interesting to either prove that the first rows of $\widetilde{A}_1$ and $\widetilde{A}_2$ are linearly independent for any $\Omega$, or to find counterexamples–namely, inclusions that are neutral in only one direction.

**Remark 2.** *According to the universal approximation theorem, for a given interface function $p(x)$, the analytic solution $u_p$ on a bounded set and $p$ on $\partial\Omega$ can be approximated by neural networks. Additionally, by the Fourier analysis, $p(x)$ can be approximated by a truncated Fourier series $p^{(n)}$. In light of Theorem 3 and Remark 1, we train using only a single linear field $H$.*

Figure 3 demonstrates the operational principles of the neural networks we designed. We use 'Univ.' to represent the universal approximation theorem, 'Fourier' for the Fourier series expansion, Thm. 1 for the analytical solution derived from the mathematical result, and 'Cred.' for credibility. We note that $u_p$ is the analytic solution to Eq. (1) associated with $p$ obtained either from $p^{(n)}$ via CoCo-PINNs or from $p_{\text{NN}}$ via classical PINNs. The credibility of $u_p$ is determined by whether $\|u_p - u_{\text{NN}}\|$ is

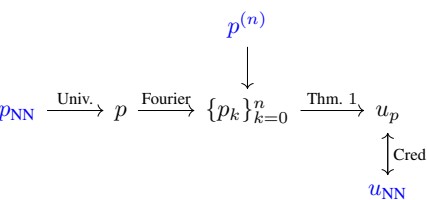

Figure 3: Credibility scheme. $u_{\text{NN}}, p_{\text{NN}}, p^{(n)}$ are trained results by PINNs.

small. Credibility is a crucial factor in the proposed PINNs' schemes for identifying neutral inclusions with imperfect boundary conditions. If the trained forward solution $u_{\text{NN}}$, obtained alongside with $p^{(n)}$ or $p_{\text{NN}}$, is close to $u_p$, we can conclude that the neural networks have successfully identified the interface function, ensuring that $\Omega$ exhibits the neutral inclusion effect. This is because $(u_{\text{NN}} - H)$ has small values in $\Omega^{\text{ext}}$ by the definition of the loss function $\mathcal{L}_{\text{Neutral}}$. In other words, if the interface function provided by neural networks makes both $\|u_p - u_{\text{NN}}\|$ and $\mathcal{L}_{\text{Neutral}}$ small, then this interface function is the desired one. However, if $\|u_p - u_{\text{NN}}\|$ is not small, it becomes unclear whether the neural networks have failed to solve the inverse problem or the forward problem.

The field of AI research is currently facing significant challenges regarding the efficacy and explainability of solutions generated by neural networks. Moreover, there is a pressing need to establish "reliability" credibility in these solutions. It is noteworthy that the trained forward solution $u_{\text{NN}}$ deviates from the analytic solution $u_p$ defined with $p_{\text{NN}}$ in several examples, particularly in cases involving complicated-shaped inclusions (see Section 4.2). This discrepancy raises concerns about the "reliability" credibility of neural networks.

## 4 EXPERIMENTS

We present the successful outcomes for designing the neutral inclusions by using the CoCo-PINNs, as well as the experiment results to verify the "reliability" in terms of credibility, consistency, and stability. *Credibility* indicates whether the trained forward solution closely approximates the analytic solution, which we assess by comparing the trained forward solution with the analytic solution derived in Choi & Lim (2024). *Consistency* focuses on whether the interface functions obtained from the classical PINNs and the CoCo-PINNs converge to the same result for re-experiments under identical environments. It's worth noting that even if the neural networks succeed in fitting the forward solution and identifying the interface parameter in a specific experiment, this success may only occur occasionally. Consistency is aimed at determining whether the training outcomes are steady or merely the result of chance, and it can be utilized as an indicator of the steadiness of the model. training model. Lastly, *stability* refers to the sensitivity of a training model, examining how the model's output changes in response to variations in environments of PDEs.

The inclusions with the shapes illustrated in Figure 4 will be used throughout this paper. Each shaped inclusion is defined by the conformal map given by Eqs. (26) to (29) in Appendix D.3. We named the shapes of the inclusions 'square', 'fish', 'kite', and 'spike'.

We introduce the quantities to validate the credibility and the neutral inclusion effect as follows:

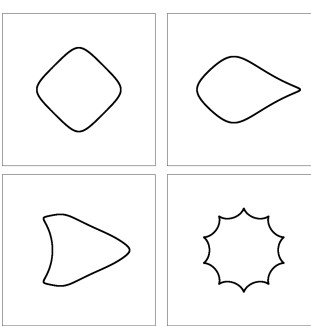

$$\|u_{\mathrm{NN}}^{\mathrm{ext}} - u_p\|_{\mathrm{Cred}} = \frac{1}{|\Omega^{\mathrm{ext}}|} \sum_{x \in \Omega^{\mathrm{ext}}} |u_{\mathrm{NN}}^{\mathrm{ext}} - u_p|^2, \quad (14)$$

$$\|u_{\mathrm{NN}}^{\mathrm{ext}} - u_p\|_{\infty} = \max_{x \in \Omega^{\mathrm{ext}}} \left\{ |u_{\mathrm{NN}}^{\mathrm{ext}} - u_p| \right\}, \quad (15)$$

$$\|u_p - H\|_{\mathrm{P\text{-}Neutral}} = \frac{1}{|\Omega^{\mathrm{ext}}|} \sum_{x \in \Omega^{\mathrm{ext}}} |u_p - H|^2, \quad (16)$$

Figure 4: The shaped inclusions: square, fish, kite, and spike.

## 4.1 NEUTRAL INCLUSION

We present experimental results demonstrating the successful achievement of the neutral inclusion effect using CoCo-PINNs. For training, we use only a single background solution, $H(x) = x_1$, and illustrate the neutral inclusion effects for three background solutions $H(x) = x_1, x_2$ and $2x_1 - x_2$; see Theorem 3 for a theoretical explanation.

Inclusions generally yield perturbations in the applied background fields. However, the domain $\Omega$, with the interface function $p(x)$ trained from CoCo-PINNs, achieves the neutral inclusion effects across all three test background fields, as shown by the level curves of the analytic solutions $u_p$ in Figure 5. All results are presented in Figure 10 of Appendix D.5.

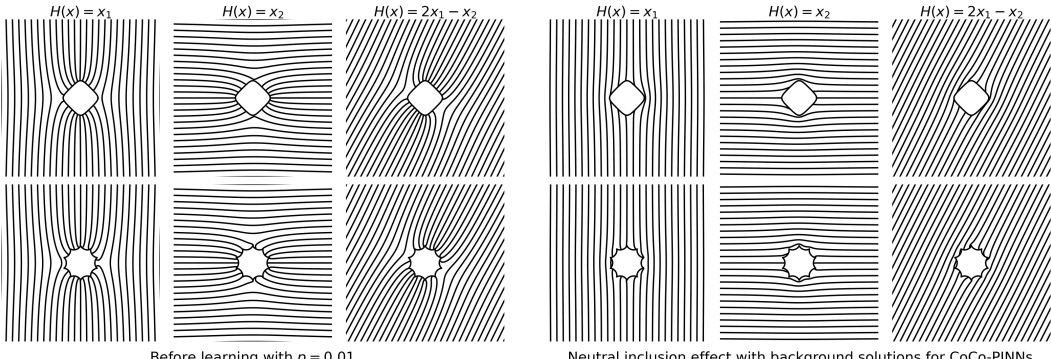

Figure 5: Neutral inclusion effect appeared after training. For the 'square' and 'spike' shaped inclusions, the interface function $p^{(n)}$ is separately trained using a single background solution $H(x) = x_1$ via CoCo-PINNs.

## 4.2 CREDIBILITY OF CLASSICAL PINNS AND COCO-PINNS

We investigate the credibility of the two methods. We examine whether the exterior part of the trained forward solution $u_{\mathrm{NN}}^{\mathrm{ext}}$ matches the analytic solution $u_p$. Recall that, we denote $u_p$ as the analytic solution when coefficients of the interface function $p$ are given by training. Once training is complete, CoCo-PINNs provide the expansion coefficients of the interface function directly. For classical PINNs, where the inter-

Table 1: Credibility results.

| | Shape | Credibility | |
|---|---|---|---|
| | | $\|u_{\mathrm{NN}}^{\mathrm{ext}} - u_p\|_{\mathrm{Cred}}$ | $\|u_{\mathrm{NN}}^{\mathrm{ext}} - u_p\|_{\infty}$ |
| Classical PINNs | square | **7.241e-04** | 2.617e-01 |
| | fish | 4.704e-04 | 1.005e-01 |
| | kite | 8.790e-03 | 5.999e-01 |
| | spike | 1.371e-02 | 5.798e-01 |
| CoCo-PINNs | square | 3.197e-03 | **9.958e-02** |
| | fish | **2.505e-04** | **8.960e-02** |
| | kite | **1.947e-03** | **2.117e-01** |
| | spike | **3.969e-03** | **2.431e-01** |

face function is represented by neural net-
works, we compute the Fourier coefficients
of $p_{\text{NN}}$. We use the Fourier series expansion up to a sufficiently high order to ensure that the difference between the neural network-designed interface function and its Fourier series is small (see Figure 9 in Appendix D.4). We experiment for four inclusion shapes in Figure 4. Detailed experiment settings are given in the Appendix D.

Table 1 demonstrates CoCo-PINNs have superior performance in the shape of 'fish', 'kite', and 'spike' compared to the classical PINNs. Although the credibility error for classical PINNs appears smaller than that for CoCo-PINNs on the 'square' shape as shown in Table 1, the trained forward solution by classical PINNs illustrates an exorbitant large deviation that does not coincide with the analytic solution derived from the inverse parameter result, as shown in Figure 6. This indicates that, despite its strong performance in minimizing the loss function, the classical PINNs approach fails to effectively function as a forward solver.

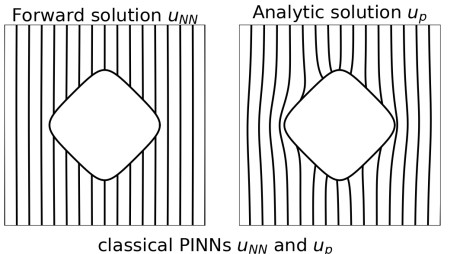 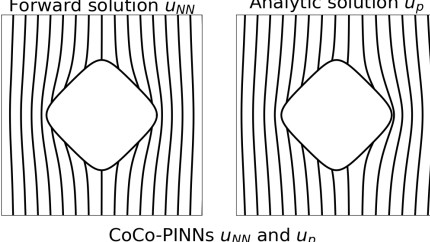

Figure 6: The trained forward solution, $u_{\text{NN}}$, and the analytic solution $u_p$ for classical PINNs and Coco-PINNs with 'square' shape. Note that $u_{\text{NN}}$ should closely resemble the analytic solution $u_p$.

### 4.3 CONSISTENCY OF CLASSICAL PINNs AND CoCo-PINNs

In this subsection, we examine whether repeated experiments consistently yield similar results. Figure 7 shows the interface functions after training the classical PINNs and CoCo-PINNs performed independently multiple times. We repetitively test 30 times under the same condition and plot the interface function pointwise along the boundary of the unit disk. The blue-dashed and red-bold lines represent the mean of the interface functions produced by classical PINNs and CoCo-PINNs, respectively, while the blue- and red-shaded regions indicate the pointwise standard deviations of the interface functions, respectively. Each column corresponds to an experiment with 'square', 'fish', 'kite', and 'spike'.

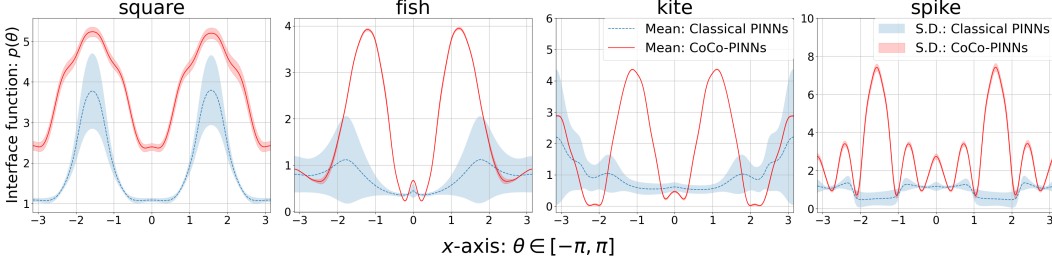

Figure 7: Consistency of interface functions in classical PINNs and CoCo-PINNs.

As shown in Figure 7, the interface functions trained by classical PINNs show inconsistency, while CoCo-PINNs produce consistent results. The precise value for the mean of the standard deviations is provided in column 3 of Table 2. Additionally, columns 4 through 6 of Table 2 present the three quantities in Eqs. (14) to (16), which show the credibility and the errors indicating the neutral inclusion effect. These results clearly demonstrate that CoCo-PINNs exhibit superior credibility compared to classical PINNs. Furthermore, CoCo-PINNs outperform classical PINNs in achieving the neutral inclusion effect, particularly for complex shapes, as verified by the quantity $\|u_p - H\|_{\text{P-Neutral}}$. In some cases, the trained forward solution $u_{\text{NN}}$ and the inverse solution $p_{\text{NN}}$ produced by classical

PINNs show a significant discrepancy, with $u_{\text{NN}}$ differing substantially from the analytic solution derived from $p_{\text{NN}}$ (see Figure 11 in Appendix D.6).

Table 2: Mean of the standard deviation of the interface function, and mean and standard deviations of errors for fitting the background fields after training. CoCo-PINNs show superior results than classical PINNs for complex shapes such as 'fish', 'kite', and 'spike'. The 'Mean of S.D.' denotes the mean of standard deviation for each point, and 'S.D.' denotes the standard deviation.

| | Shape | Interface function | $\|u_{\text{NN}}^{\text{ext}} - u_p\|_{\text{Cred}}$ | | $\|u_{\text{NN}}^{\text{ext}} - u_p\|_{\infty}$ | | $\|u_p - H\|_{\text{P-Neutral}}$ | |
|---|---|---|---|---|---|---|---|---|
| | | Mean of S.D. | Mean | S.D. | Mean | S.D. | Mean | S.D. |
| Classical PINNs | square | 2.614e-01 | **1.071e-03** | **1.348e-04** | 3.185e-01 | 2.601e-02 | **1.102e-03** | **1.043e-04** |
| | fish | 4.011e-01 | 1.685e-03 | 1.375e-03 | 1.223e-01 | 1.606e-02 | 2.220e-03 | 1.895e-03 |
| | kite | 5.358e-01 | 1.779e-02 | 2.412e-02 | 1.173e+00 | 1.082e+00 | 1.676e-02 | 2.454e-02 |
| | spike | 2.554e-01 | 1.094e-02 | 2.511e-03 | 5.188e-01 | 7.791e-02 | 1.636e-03 | 1.048e-03 |
| CoCo-PINNs | square | **1.357e-01** | 3.190e-03 | 1.515e-04 | **1.014e-01** | **2.152e-03** | 5.045e-03 | 2.443e-04 |
| | fish | **3.264e-02** | **2.482e-04** | **1.554e-05** | **9.028e-02** | **5.617e-03** | **4.065e-04** | **6.261e-06** |
| | kite | **2.258e-02** | **1.314e-03** | **4.500e-04** | **1.800e-01** | **2.625e-02** | **4.267e-04** | **1.990e-05** |
| | spike | **1.432e-01** | **3.512e-03** | **1.505e-04** | **2.336e-01** | **3.575e-03** | **1.367e-03** | **3.073e-04** |

## 4.4 STABILITY OF CLASSICAL PINNs AND COCO-PINNs

In this subsection, we assess the stability of the interface function along with the change of environments of PDEs. Since both classical PINNs and CoCo-PINNs are trained for a fixed domain $\Omega$ and background field $H$, we focus on stability with respect to different conductivities $\sigma_c$. In Figure 8, we present experiments results obtained for $\sigma_c = 3, 4, 5, 6, 7$ and $\sigma_m = 1$, where the inclusion shapes are 'square', 'fish', 'kite', and 'spike'. Recall that the ill-posed nature of inverse problems can lead to significant instability, causing the inverse solution to exhibit large deviations in response to environmental changes or re-experimentation. Classical PINNs for neutral inclusions with imperfect conditions represent such an unstable case, as demonstrated in Figure 8. In contrast, our CoCo-PINNs are stable for repeated experiments, and we confirmed that CoCo-PINNs are stable for slightly changed environments. Table 3 provides the mean of standard deviations for all experiments used in Figure 8. As shown in Table 3, the CoCo-PINNs are exceedingly stable for consistency and stability than classical PINNs.

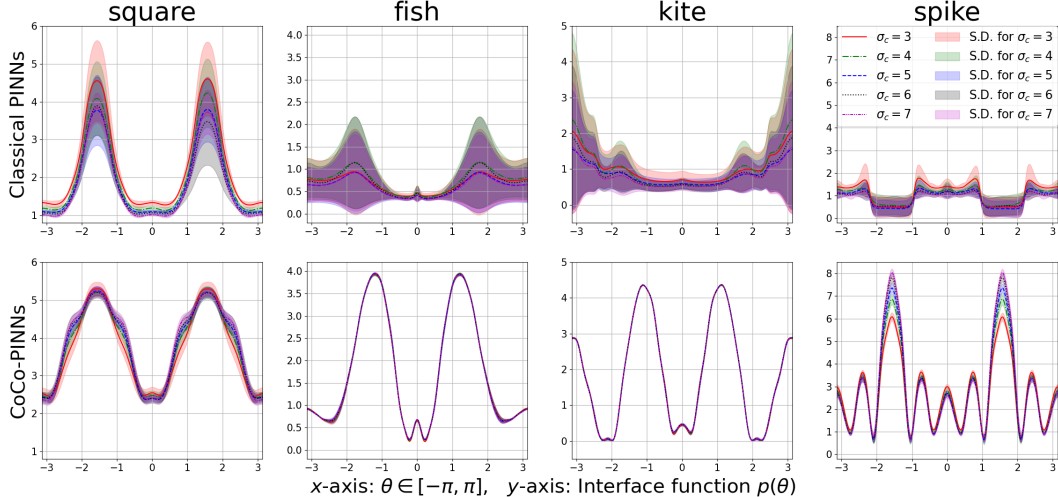

Figure 8: The first and second rows present the stability of interface functions from classical PINNs and CoCo-PINNs, respectively. The precise value for the mean of standard deviations is in Table 3.

Table 3: Mean of standard deviations of the interface function for stability experiments.

| | Shape | Conductivities for interior | | | | |
|---|---|---|---|---|---|---|
| | | $\sigma_c = 3$ | $\sigma_c = 4$ | $\sigma_c = 5$ | $\sigma_c = 6$ | $\sigma_c = 7$ |
| Classical PINNs | square | 3.060e-01 | 2.784e-01 | 2.614e-01 | 2.840e-01 | **1.289e-01** |
| | fish | 4.421e-01 | 4.416e-01 | 4.056e-01 | 4.218e-01 | 3.734e-01 |
| | kite | 6.282e-01 | 6.062e-01 | 4.703e-01 | 4.722e-01 | 4.597e-01 |
| | spike | 3.505e-01 | 2.855e-01 | 2.469e-01 | 2.588e-01 | 2.420e-01 |
| CoCo-PINNs | square | **2.082e-01** | **1.565e-01** | **1.357e-01** | **1.775e-01** | 1.457e-01 |
| | fish | **2.841e-02** | **3.077e-02** | **2.760e-02** | **2.611e-02** | **3.105e-02** |
| | kite | **1.874e-02** | **2.346e-02** | **2.095e-02** | **2.325e-02** | **2.420e-02** |
| | spike | **1.142e-01** | **1.709e-01** | **1.313e-01** | **1.468e-01** | **1.513e-01** |

## 5 CONCLUSION

We focus on the inverse problem of identifying an imperfect function that makes a given simply connected inclusion a neutral inclusion. We introduce a novel approach of Conformal mapping Coordinates Physics-Informed Neural Networks (CoCo-PINNs) based on complex analysis and PDEs. Our proposed approach of CoCo-PINNs successively and simultaneously solves the forward and inverse problem much more effectively than the classical PINNs approach. While the classical PINNs approach may occasionally demonstrate success in finding an imperfect function with a strong neutral inclusion effect, the reliability of this performance remains uncertain. In contrast, CoCo-PINNs present high credibility, consistency, and stability, with the additional advantage of being explainable through analytical results. The potential applications of this method extend to analyzing and manipulating the interaction of embedded inhomogeneities and surrounding media, such as finding inclusions having uniform fields in their interiors. Several questions remain, including the generalization to multiple inclusions and three-dimensional problems, as well as proving the existence of an interface function that achieves neutrality.

ETHICS STATEMENT

This work explores the development of machine learning approaches for advancing the theory of composite materials, particularly addressing the problem of neutral inclusions, with a focus on positive applications such as designing reinforced or embedded composites that preserve the original stress field of the material without inclusions and avoid stress concentrations. We emphasize that this research is conducted to promote constructive and ethical applications.

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

# A  NOTATIONS

Table 4 provides the list of notations used throughout the paper.

Table 4: Notations

| Notation | Meaning |
| --- | --- |
| $H(x)$ | background fields |
| $p(x)$ | interface function |
| $p_{\text{NN}}$ | Inverse solution via classical PINNs |
| $p^{(n)}$ | Truncated representation of interface function & Inverse solution via CoCo-PINNs |
| $u_p$ | Analytic solution with given interface function |
| $u_{\text{NN}}$ | Trained forward solution |

# B  GEOMETRIC FUNCTION THEORY

Geometric function theory is the research area of mathematics with the corresponding geometric properties of analytic functions. One remarkable result is the Riemann mapping theorem. We briefly introduce this theorem with related results.

## B.1  RIEMANN MAPPING, FABER POLYNOMIALS, AND GRUNSKY COEFFICIENTS

A connected open set in the complex plane is called a domain. We say that a domain $\Omega$ is simply connected if its complement $\mathbb{C} \setminus \Omega$ is connected.

**Theorem 4** (Riemann mapping theorem). *If $\Omega \subsetneq \mathbb{C}$ is a nonempty simply connected domain, then there exists a conformal map from the unit ball $B = \{z \in \mathbb{C} : |z| < 1\}$ onto $\Omega$.*

We assume that $\Omega \subsetneq \mathbb{C}$ is a nonempty simply connected bounded domain. Then, by the Riemann mapping theorem, there exists a unique $\gamma > 0$ and conformal mapping $\Psi$ from $D = \{w \in \mathbb{C} : |w| > \gamma\}$ onto $\mathbb{C} \setminus \overline{\Omega}$ such that $\Psi(\infty) = \infty$, $\Psi'(\infty) = 1$, and

$$\Psi(w) = w + a_0 + \frac{a_1}{w} + \frac{a_2}{w^2} + \cdots. \tag{17}$$

The quantity $\gamma$ in Eq. (17) is called the conformal radius of $\Omega$. One can obtain Eq. (17) by using Theorem 4, the power series expansion of an analytic function and its reflection with respect to a circle; we refer to for instance Pommerenke (1992, Chapter 1.2) for the derivation.

We further assume that $\Omega$ has an analytic boundary, that is, $\Psi$ can be conformally extended to $\{w \in \mathbb{C} : |w| > \gamma - \epsilon\}$ for some $\epsilon > 0$.

The exterior conformal mapping $\Psi$ in Eq. (17) defines the Faber polynomials $\{F_m\}_{m=1}^{\infty}$ by the relation

$$\frac{\Psi'(w)}{\Psi(w) - z} = \sum_{m=0}^{\infty} \frac{F_m(z)}{w^{m+1}}, \quad z \in \overline{\Omega},\ |w| > \gamma. \tag{18}$$

The Faber polynomials $\{F_m\}$ are monic polynomials of degree $m$, and their coefficients are uniquely determined by the coefficients $\{a_n\}_{n=0}^m$ of $\Psi$. In particular, one can determine $F_m$ by the following recursive relation:

$$F_{m+1}(z) = zF_m(z) - ma_m \sum_{n=0}^{m} a_n F_{m-n}(z),\ m \geq 0. \tag{19}$$

In particular, $F_1(z) = z - a_0$. A core feature of the Faber polynomials is that $F_m(\Psi(w))$ has only a single positive-order term $w^m$. In other words, we have

$$F_m(\Psi(w)) = w^m + \sum_{n=1}^{\infty} c_{mn} w^{-n}, \qquad |w| > \gamma,$$

where $c_{mn}$ are known as the Grunsky coefficient.

**Remark 3.** *The Faber polynomial forms a basis for complex analytic functions in $\Omega$. This means that an analytic function $v$ in $\Omega$ can be expanded into $F_m$ as*

$$v(z) = \sum_{n=0}^{\infty} b_n F_n(z), \quad z \in \Omega,$$

*for some complex coefficients $b_n$.*

## C PROOF OF MAIN THEOREM

As the Faber polynomials form a basis for complex analytic functions (see remarked in Remark 3), the background field $H$ is an entire harmonic so that it is the real part of an entire analytic function. Hence, $H$ satisfies

$$H(z) = \sum_{m=1}^{\infty} \Re[\alpha_m F_m(z)]$$

for some complex coefficients $\{\alpha_m\}$. Choi & Lim (2024) showed that, for some $\delta > 0$, the solution $u$ to the Eq. (1) admits the expression

$$u(z) = \begin{cases} \Re\left[\sum_{m=1}^{\infty}\sum_{n=1}^{\infty} \beta_{mn} F_n(z)\right] & \text{for } \rho \in [\rho_0 - \delta, \rho_0], \\ \Re\left[\sum_{m=1}^{\infty} \alpha_m F_m(z) + \sum_{m=1}^{\infty}\sum_{n=1}^{\infty} s_{mn} w^{-n}\right] & \text{for } \rho > \rho_0 \end{cases}$$

$$= \begin{cases} \Re\left[\sum_{m=1}^{\infty}\sum_{n=1}^{\infty} \beta_{mn} w^n + \sum_{m=1}^{\infty}\sum_{n=1}^{\infty}\sum_{l=1}^{\infty} \beta_{ml} c_{ln} w^{-n}\right] & \text{for } \rho \in [\rho_0 - \delta, \rho_0], \\ \Re\left[\sum_{m=1}^{\infty} \alpha_m w^m + \sum_{m=1}^{\infty}\sum_{n=1}^{\infty} (\alpha_m c_{mn} + s_{mn}) w^{-n}\right] & \text{for } \rho > \rho_0, \end{cases}$$

where the coefficients $\beta_{mn}$ and $s_{mn}$ are depending on $H$, $\sigma$ and $p$. Recall that $c_{mn}$ are Grunsky coefficients.

We define semi-infinite matrices:

$$\boldsymbol{\alpha} = \{\alpha_m \delta_{mn}\}_{m,n\geq 1}, \ \boldsymbol{\beta} = \{\beta_{mn}\}_{m,n\geq 1}, \ \boldsymbol{s} = \{s_{mn}\}_{m,n\geq 1}, \tag{20}$$

$$\mathcal{N} = \{n\delta_{mn}\}_{m,n\geq 1}, \ \gamma^{\tau\mathcal{N}} = \{\gamma^{\tau n}\delta_{mn}\}_{m,n\geq 1}, \ C = \{c_{mn}\}_{m,n\geq 1}, \tag{21}$$

where $\delta_{mn}$ is the Kronecker delta function, and $\tau \in \mathbb{R}$.

Define $\Psi(\rho, \theta) = \Psi(e^{\rho + i\theta})$ and denote the scale factor as $h$, that is,

$$h(\rho, \theta) = \left|\frac{\partial \Psi}{\partial \rho}\right| = \left|\frac{\partial \Psi}{\partial \theta}\right| = e^{\rho}|\Psi'|.$$

For $|w| = \gamma$, we have $w = e^{\rho_0 + i\theta}$ for $\theta \in [0, 2\pi)$. We consider the Fourier series of $h(\rho_0, \theta)p(\Psi(w))$ in $\theta$ :

$$h(\rho_0, \theta)p(\Psi(w)) = \sum_{n=-\infty}^{\infty} p_n w^n, \quad |w| = \gamma,$$

where $p_n \gamma^n$ is the Fourier coefficients. As $p$ is a real-valued function, one can show that $p_{-n} = \overline{p_n}\gamma^{2n}$ for each $n \in \mathbb{Z}$ and, hence,

$$h(\rho_0, \theta)p(\Psi(w)) = p_0 + p_1 w + \overline{p_1}\gamma^2 w^{-1} + p_2 w^2 + \overline{p_2}\gamma^4 w^{-2} + \cdots, \quad |w| = \gamma. \tag{22}$$

We denote

$$P^+ = \{p_{mn}^+\}_{m,n\geq 1} \quad \text{with} \quad p_{mn}^+ = p_{m+n}\gamma^{m+n},$$

$$P^- = \{p_{mn}^-\}_{m,n\geq 1} \quad \text{with} \quad p_{mn}^- = p_{m-n}\gamma^{m-n}.$$

If $m < n$, we can use $p_{m-n} = \overline{p_{-m+n}}\gamma^{2(-m+n)}$.

Recall that $\boldsymbol{\alpha}$ is given by the background field, and $C$ is determined by $\Omega$. Choi & Lim (2024) showed that

$$\boldsymbol{\alpha}A_1 + \overline{\boldsymbol{\alpha}}\,\overline{A_2} + \overline{\boldsymbol{s}}\,\overline{B_1} + \boldsymbol{s}B_2 = 0 \tag{23}$$

and, using this relation, derived the expression of $\boldsymbol{s}$ in terms of $p$, $\sigma$, and $H$ as follows:

$$\boldsymbol{s} = - \left[ \boldsymbol{\alpha}\left(A_1 - A_2\overline{B_2}^{-1}\overline{B_1}\right) + \overline{\boldsymbol{\alpha}}\left(\overline{A_2} - \overline{A_1 B_2}^{-1}\overline{B_1}\right) \right] \left(B_2 - B_1\overline{B_2}^{-1}\overline{B_1}\right)^{-1}, \tag{24}$$

where

$$\begin{cases}
A_1 &= (\sigma_c - \sigma_m)\gamma^{\mathcal{N}}\overline{P^+}\gamma^{\mathcal{N}} + (\sigma_c - \sigma_m)C\gamma^{-\mathcal{N}}P^-\gamma^{\mathcal{N}} + \sigma_c\sigma_m C\mathcal{N}, \\[2mm]
A_2 &= (\sigma_c - \sigma_m)\gamma^{\mathcal{N}}\overline{P^-}\gamma^{\mathcal{N}} + (\sigma_c - \sigma_m)C\gamma^{-\mathcal{N}}P^+\gamma^{\mathcal{N}} - \sigma_c\sigma_m\gamma^{2\mathcal{N}}\mathcal{N}, \\[2mm]
B_1 &= \left[(\sigma_c - \sigma_m)I + 2\sigma_m(I - \gamma^{-2\mathcal{N}}\overline{C}\gamma^{-2\mathcal{N}}C)^{-1}\right]\gamma^{-\mathcal{N}}P^+\gamma^{\mathcal{N}} \\[2mm]
&\quad + 2\sigma_m(I - \gamma^{-2\mathcal{N}}\overline{C}\gamma^{-2\mathcal{N}}C)^{-1}\gamma^{-2\mathcal{N}}\overline{C}\gamma^{-\mathcal{N}}\overline{P^-}\gamma^{\mathcal{N}}, \\[2mm]
B_2 &= \left[(\sigma_c - \sigma_m)I + 2\sigma_m(I - \gamma^{-2\mathcal{N}}\overline{C}\gamma^{-2\mathcal{N}}C)^{-1}\right]\gamma^{-\mathcal{N}}P^-\gamma^{\mathcal{N}} \\[2mm]
&\quad + 2\sigma_m(I - \gamma^{-2\mathcal{N}}\overline{C}\gamma^{-2\mathcal{N}}C)^{-1}\gamma^{-2\mathcal{N}}\overline{C}\gamma^{-\mathcal{N}}\overline{P^+}\gamma^{\mathcal{N}} + \sigma_c\sigma_m\mathcal{N}.
\end{cases}$$

We set

$$\widetilde{A}_1 = \left(A_1 - A_2\overline{B_2}^{-1}\overline{B_1}\right)\left(B_2 - B_1\overline{B_2}^{-1}\overline{B_1}\right)^{-1},$$

$$\widetilde{A}_2 = \left(\overline{A_2} - \overline{A_1 B_2}^{-1}\overline{B_1}\right)\left(B_2 - B_1\overline{B_2}^{-1}\overline{B_1}\right)^{-1}.$$

**Proof of Theorem 3.** The neutral inclusion means that $\Re(\boldsymbol{s}\boldsymbol{w}^{-\mathcal{N}}) = 0$ where $\boldsymbol{w}^{-\mathcal{N}} = \{w^{-n}\}_{n\geq 1}$ for all $|w| > \gamma$. This implies that $\boldsymbol{s} = 0$. In other words, $\boldsymbol{\alpha}\widetilde{A}_1 + \overline{\boldsymbol{\alpha}}\widetilde{A}_2 = 0$.

If the background field is $H(x) = ax_1 + bx_2$, $H(z) = \Re\left[\sum_{m=1}^{\infty}\alpha_m F_m(z)\right]$ with $\alpha_1 = a - ib$ and $\alpha_2 = \alpha_3 = \cdots = 0$ so that the matrix $\boldsymbol{\alpha}$ is

$$\boldsymbol{\alpha} = \begin{bmatrix} \alpha_1 & 0 & \cdots \\ 0 & 0 & \ddots \\ \vdots & \ddots & \ddots \end{bmatrix} \text{ with } \alpha_1 \in \mathbb{C}, \quad \alpha_1 = a - ib. \tag{25}$$

Hence, $\boldsymbol{s}$ has nonzero entries only in its first row. Hence, $\Omega$ is neutral to the linear field $H$ if and only if $-\alpha_1 \operatorname{row}_1[\widetilde{A}_1] - \overline{\alpha_1}\operatorname{row}_1[\widetilde{A}_2] = 0$. By the assumption of the theorem, the first rows of $\operatorname{row}_1[\widetilde{A}_1]$ and $\operatorname{row}_1[\widetilde{A}_2]$ are linearly independent. Hence, $\operatorname{row}_1[\widetilde{A}_1] = \operatorname{row}_1[\widetilde{A}_2] = 0$. This implies that $\Omega$ is neutral to linear fields $H$ of all directions. $\qquad\square$

# D  EXPERIMENTAL DETAILS

## D.1  COLLOCATION POINTS

We denote the set of collocation points as

$$\Omega^{\text{int}}, \ \Omega^{\text{ext}}, \ \partial\Omega, \ \partial\Omega^+, \ \partial\Omega^-,$$

for interior, exterior, boundary, and the limit to the boundary from interior and exterior components, respectively. We define as follows: $\Omega^{\text{ext}}$ and $\partial\Omega$ are the images of the exterior conformal map $\Psi(w)$ under the uniform grid of its restricted domain, and the boundary of the disk with radius $\gamma = e^{\rho_0}$ by

$$\Omega^{\text{ext}} = \left\{ \Psi(w) : w = e^{\rho+i\theta}, \ (\rho,\theta) \in (\rho_0, L] \times (0, 2\pi] \right\},$$

$$\partial\Omega = \left\{ \Psi(w) : w = e^{\rho_0+i\theta}, \ \theta \in (0, 2\pi] \right\},$$

for some fixed $L > \gamma$. The limits to the boundary $\partial\Omega^{\pm}$ from exterior and interior are defined by

$$\partial\Omega^{\pm} = \{z \pm \delta N(z) : z \in \partial\Omega\}$$

with a small $\delta > 0$ and a unit normal vector $N$ to the boundary $\partial\Omega$. To select the interior points, we recall that $z \in \partial\Omega$ can be represented as $z = \Psi(e^{\rho_0 + i\theta})$. Now, we fix the angle $\theta$ and divide the radius uniformly. In other words, we collect $x \in \Omega$ such that $\arg(x) = \arg(z)$ and $|x| < |z|$, that is,

$$\Omega^{\text{int}} = \{x \in \Omega : |x| < |z|, \ \arg(x) = \arg(z), \ z \in \partial\Omega\}.$$

### D.2 Experimental setup and parameters

In common settings for experiments, we set $\sigma_m = 1, \sigma_m = 5$ and $\sigma_c = 3, 4, 6, 7$ are also used. The conformal radius $\gamma = 1$ and $L = 5$. which determines the domain of the conformal map, and sample points are located in the square box of $[-5, 5]^2$. We generated collocation points $|\Omega^{\text{ext}}| = 21,808, |\Omega^{\text{int}}| = 6,000$, and $|\partial\Omega| = 6,000$ with a single epoch so that collocation points are all fixed. For the neural networks' architecture, we use 4 hidden layers with a width of 20 and the Tanh activation due to its smoothness. We use Adam optimizers under 25,000 iterations adjusting learning rates **lr pinn**: all types of neural networks and **lr inv**: for interface (inverse) parameters decaying with $\eta$ per 1000 iterations along the learning rate schedulers. We fix both learning rates as $10^{-3}$ and $\eta = 0.7$. We use the CoCo-PINNs with $p = p^{(20)}$ which takes a real number $p_0$ and complex numbers $\{p_i\}_{i=1}^{20}$ as inverse parameters. To verify the classical PINNs' result, we use the Fourier fitting with an order of 20. The Fourier fitting is explained in Appendix D.4. Note that using too high order makes singularities, and hence, it represents awful credibility. We set the initial value of the interface function to 5 for the square and spike shape to make it easier to satisfy the condition of being a positive function.

**Remark on the environments**     Traditionally, hyperparameter tuning is important to acquire the PINNs' performance. Since balancing them is much more complicated due to the difficulty of the PDEs, we need to choose the appropriate values to enhance PINNs. Here are the brief guidelines for our settings: 1) When we balance the number of collocation points, although the exterior domain is much larger than the interior, fitting the background fields on the exterior domain has been designed to be less complex. So we set 4 times of interior points. Also, the boundary effect is essential for neutral inclusion, we put more points on the boundary same with the interior's points. 2) Re-sampling collocation points can lead to the uncertainty of the interface function. If we sample again for each epoch, PINNs are hardly adapted for new sample points, especially for the boundary condition loss, which is quite important for the neutral inclusion effects. It is also from the experimental results that the results of re-sampling are worse than those of fixed-sampling. Hence, we fix the points and proceed to a single epoch. 3) To control the learning rates adaptively, we use the Adam optimizers so that we can handle the sensitivity of Fourier coefficients much better. Since Adam has faster convergence and robustness, they can adjust the learning rates appropriately during the training.

The following algorithm explains the progress of the CoCo-PINNs and classical PINNs.

---

**Algorithm 1** Generation method of the perturbed field from the interface function

---

**Input:** Background fields: $H(x, y)$; Interface function: $p = p^{(n)}$ or $p_{\text{NN}}$

1: **if** $p = p^{(n)}$ **then**
2:     **[Initialization]**: Utilizing the mathematical results
3:     Training
4:     $p^{(n)}$ gives us the coefficients $\{p_k\}_{k=1}^n$ directly.
5: **else if** $p = p_{\text{NN}}$ **then**
6:     **[Initialization]**: None
7:     $p_{\text{NN}}(w)$ with given sample points $w \in \partial D$.
8:     Training
9:     Use the Fourier fitting to attain the coefficients $\{p_k\}_{k=1}^n$.
10: **end if**
11: S = Numeric$[\{p_k\}_{k=1}^n]$
12: $u_p = \Re[H + Sw] \cdots\cdots$ Eq. (3).
**Output:** $u_p$

---

### D.3 CONFORMAL MAPS FOR VARIOUS SHAPES

The shapes shown in Figure 4 are defined by the conformal map $\Psi(w) : \{w \in \mathbb{C} : |w| > 1\} \to \mathbb{C} \backslash \overline{\Omega}$ as follows:

$$\Psi(w) = w + \frac{1}{10w^3}, \tag{26}$$

$$\Psi(w) = w + \frac{1}{4w} + \frac{1}{8w^2} + \frac{1}{10w^3}, \tag{27}$$

$$\Psi(w) = w + \frac{1}{10w} + \frac{1}{4w^2} - \frac{1}{20w^3} + \frac{1}{20w^4} - \frac{1}{25w^5} + \frac{1}{50w^6}, \tag{28}$$

$$\Psi(w) = w - \frac{1}{10w^9}. \tag{29}$$

The Eqs. (26) to (29) present 'square', 'fish', 'kite', and 'spike', respectively.

### D.4 FOURIER FITTINGS

In order to ascertain whether the trained forward solution is true or not, it is necessary to identify the Fourier series that is sufficiently similar to the original interface function and, hence, achieve the real analytic solution. We utilize the Fourier series approximation for each interface function. Figure 9 presents the difference between the interface function and the Fourier series we used. We denote $p_F$ as the Fourier series corresponding to the $p_{NN}$. Given that $p_{NN}$ is sufficiently close to $p_F$, it is reasonable to utilize the analytic solution obtained by $p_F$ in order to ascertain the credibility of the classical PINNs results.

The relative $L^2$ error of the neural network-designed interface function and its Fourier series formula is given by

$$\frac{\|p_{NN} - p_F\|_{L^2(\partial\Omega)}}{\|p_{NN}\|_{L^2(\partial\Omega)}}.$$

The relative errors and the Fourier fittings for each shape are given by Table 5 and Fig. 9, respectively.

Table 5: The error of Fourier fitting

| square | fish | kite | spike |
|---|---|---|---|
| 5.013e-07 | 2.915e-07 | 1.729e-06 | 2.443e-06 |

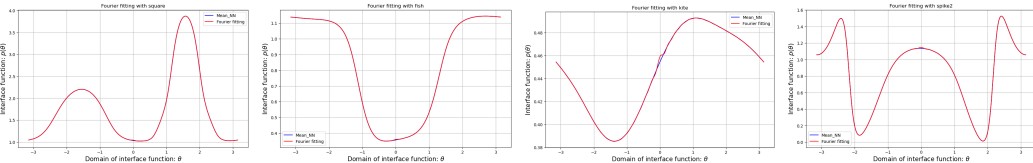

Figure 9: Fourier fitting the interface function for each domain

### D.5 NEUTRAL INCLUSION EFFECT FOR ARBITRARY FIELDS

In this subsection, we present the neutral inclusion effect of shapes and different fields $H(x_1, x_2) = x_1, x_2$, and $2x_1 - x_2$. The neutral inclusion effects for one random experiment results are given in Table 6.

After many fair experiments with both CoCo-PINNs and classical PINNs, we concluded that CoCo-PINNs were superior. After that, we tested the neutral inclusion effect for each shape by utilizing the CoCo-PINNs. Figure 10 presents the results. In the case of unit circle inclusion, exact neutral inclusion appeared. The shape we used may have no interface functions that make exact neutral inclusions. Notwithstanding, the CoCo-PINNs results for the neutral inclusion effect are, to some extent, satisfactory.

Table 6: Errors for fitting the background fields after neutral inclusion

| | Shape | $\|u_p - H\|_{\text{P-Neutral}}$ with | | |
|---|---|---|---|---|
| | | $H(x) = x_1$ | $H(x) = x_2$ | $H(x) = 2x_1 - x_2$ |
| CoCo-PINNs | square | 5.004e-03 | 8.164e-03 | 2.801e-02 |
| | fish | 4.050e-04 | 2.694e-04 | 1.868e-03 |
| | kite | 4.358e-04 | 1.654e-04 | 1.928e-03 |
| | spike | 2.230e-03 | 5.847e-03 | 1.498e-02 |

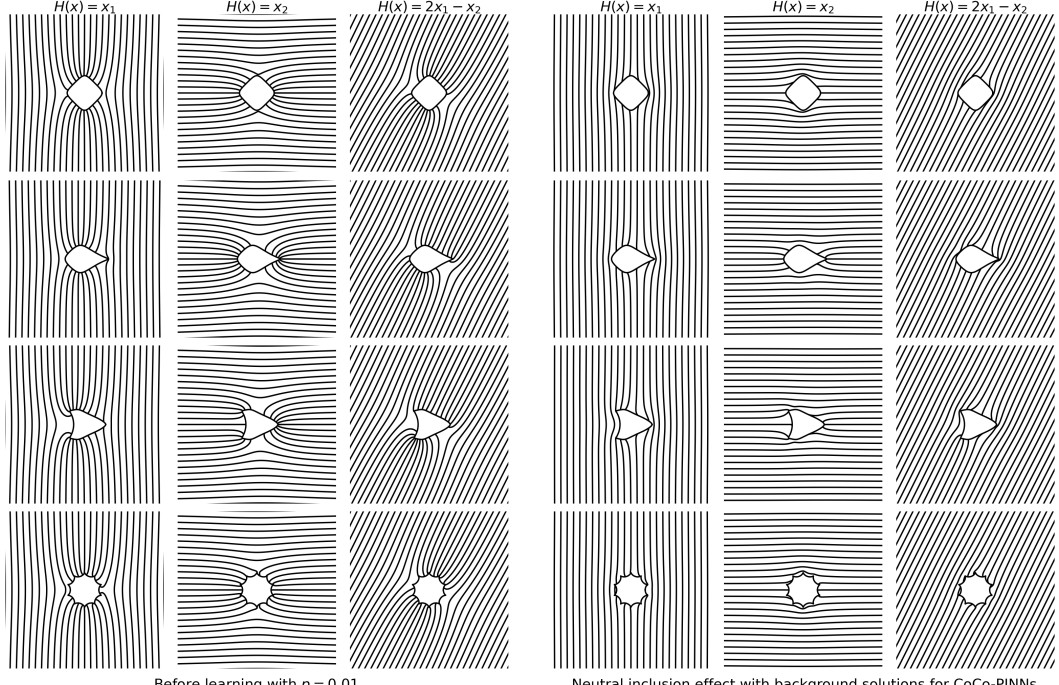

Figure 10: Neutral inclusion effects for various shapes and background fields.

### D.6 ILLUSTRATIONS FOR CREDIBILITY

All experiments described in Table 1 are illustrated in Figures 11 and 12, by utilizing CoCo-PINNs and classical PINNs, respectively. We illustrate the pairs of

$$\left( u_{\text{NN}}, u_p, \frac{|u_{\text{NN}}^{\text{ext}} - u_p|^2}{|\Omega^{\text{ext}}|}, \frac{|u_{\text{NN}}^{\text{ext}} - H|^2}{|\Omega^{\text{ext}}|} \right)$$

for each shape in Figure 4.

As shown Figures 11 and 12, the trained forward solutions by CoCo-PINNs $u_{\text{NN}}$ are more similar to analytic solution $u_p$ using the interface parameter given by the CoCo-PINNs' training results than classical PINNs' one.

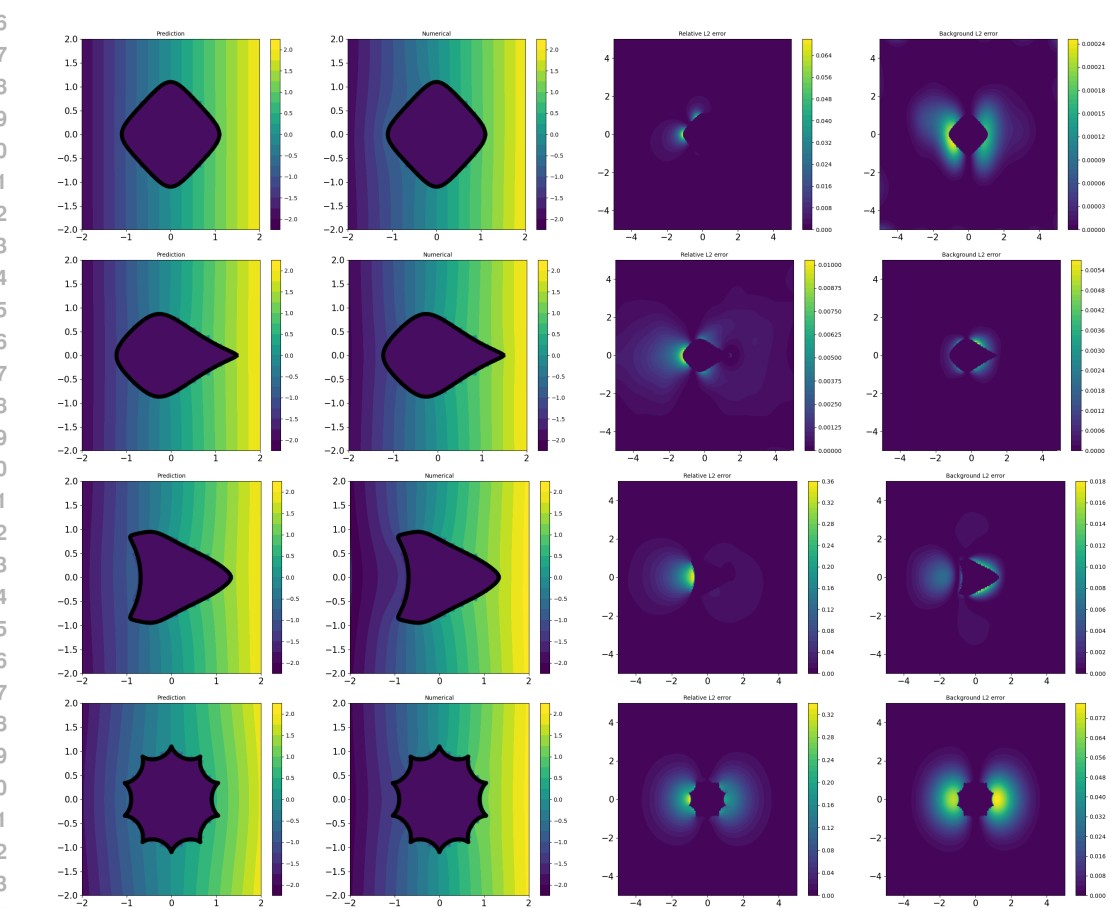

Figure 11: Experiment results by using the classical PINNs

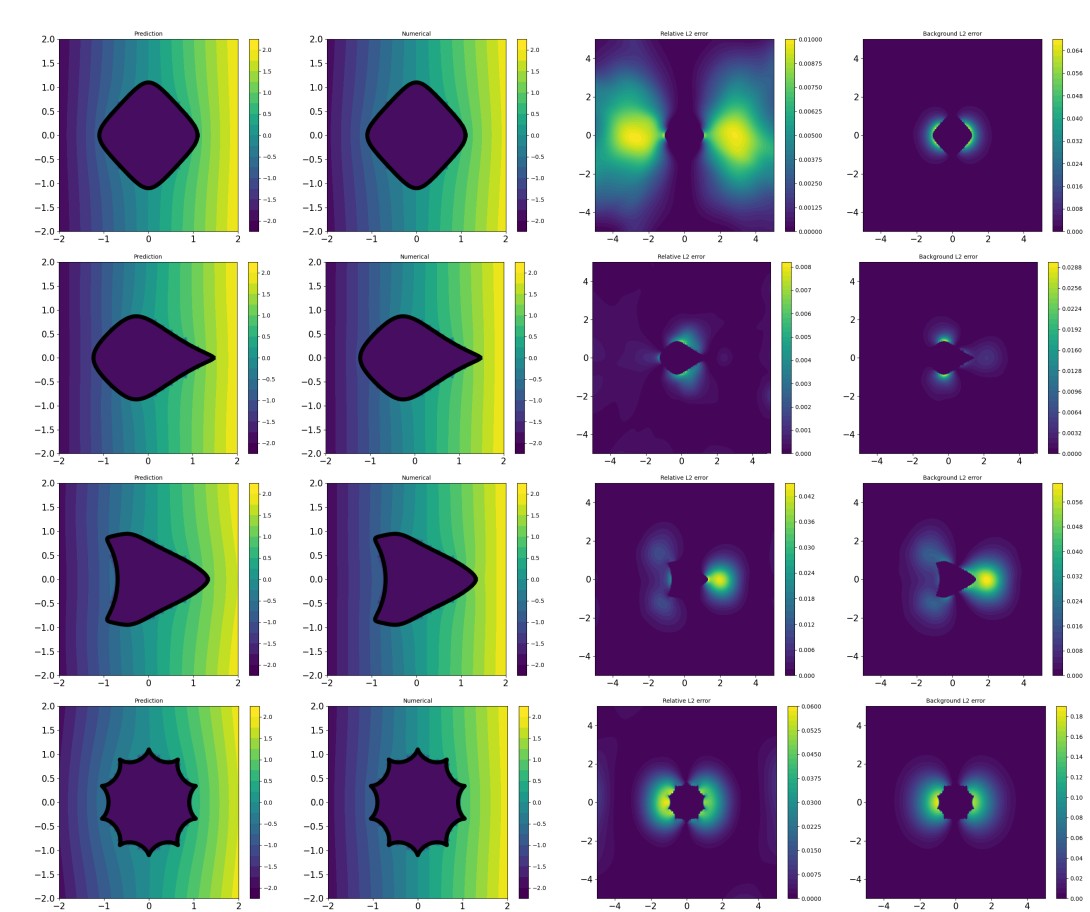

Figure 12: Experiment results by using the CoCo-PINNs

