# OpenReview forum: "Conformal mapping Coordinates Physics-Informed Neural Networks (CoCo-PINNs): learning neural networks for designing neutral inclusions"
_ICLR.cc/2025/Conference — Submitted to ICLR 2025_

### Official Review · Reviewer_JNhw · 2024-10-25

**Soundness:** 3
**Presentation:** 3
**Contribution:** 3
**Rating:** 5
**Confidence:** 3

**Summary:**

The paper introduces a new framework called CoCo-PINNs, which combines mathematical analytical methods with the Physics-Informed Neural Networks (PINNs) approach to solve the neutral inclusion problem using neural networks. The authors point out that traditional PINNs using neural networks face challenges with this problem and propose representing the interface function using series expansions instead of neural networks.

**Strengths:**

1. Numerical results demonstrate that the method offers improved credibility, consistency, and stability compared to classical PINNs.

2. Utilizing series expansions to represent inverse parameters not only improves the accuracy of the solutions but also provides valuable analytical insights.

3. An additional advantage of the method is its ability to operate effectively without the need for training data to solve the neutral inclusion problem.

**Weaknesses:**

1. The paper primarily focuses on two-dimensional problems. The lack of extensive validation in higher dimensions may limit the immediate applicability of the proposed method to more complex real-world situations. In my view, the method's reliance on the Riemann mapping theorem in complex space restricts it to 2D problems, making its extension to 3D cases unfeasible.

2. Although the paper claims improved performance over classical PINNs, it would benefit from a more detailed comparative analysis with other state-of-the-art methods. As the reviewer is not an expert in this field, the introduction's overview of related work is appreciated. However, it seems to reference many effective methods without explicitly including a discussion of PINNs.

3.  Some sections lack clarity. Please refer to the following Questions part for further elaboration.

**Questions:**

1. Line 170,The method requires the explicit formulation of the mapping $\Phi$, correct? Are we able to obtain this explicit formulation for any interface, or are there specific conditions that must be satisfied?

2. Equation 1,Could the paper clarify which of the following terms are known and which are unknown and need to be solved? I also suggest moving the condition in line 166, stating that $u$ and $H$ are equal in the exterior region, to an appropriate position at the beginning. Additionally, it would be helpful to provide several examples of $H$.

3. Theorem 1 states that \(u - H\) can be represented by a series. Why, then, is the interface function \(p\), rather than \(u\), parametrized by a series? Is there a connection between Theorem 1 and the proposed method, or is Theorem 1 only used to assess the performance of the PINNs?

4. For Theorem 2: what does it mean for a domain $\Omega$ to be neutral? Could the paper provide a detailed and rigorous definition?

5. In Section 4, all considered interfaces are defined by the conformal map. Is this a necessary condition for the proposed method? It would be helpful to clarify this point.

6. The considered $H$ functions are all 2D linear functions. Is this a necessary condition for the proposed method? It would be helpful to clarify this point.

---

> ### Author Response · Authors · 2024-11-20
> **Response to Reviewer JNhw**
>
> We thank the intuitive comments from the reviewer. Here are some clarifications from us.
> ### W1 (weakness). 3-dimensional approach
> We thank the reviewer for the valuable comment. As you mentioned, our approach is based on the conformal mapping coordinates. In 3-dimensions, if explicit coordinate systems can be defined near the boundary of a given inclusion, simple examples are balls and ellipsoids, we can apply the proposed approach.
>
> ### W2. State-of-arts PINNs
> Thank you for your insightful comments regarding state-of-the-art models that build upon classical PINNs. As mentioned in line 47 of the Introduction section, numerous studies have modified PINNs to address specific inverse problems. For example, Pokkunuru et al. (2023) utilized a Bayesian approach to design the loss function, Guo et al. (2022) employed Monte Carlo approximations to compute fractional derivatives, Xu et al. (2023) adopted a multi-task learning approach to weight losses and presented forward-inverse problem-combined neural networks, and Yuan et al. (2022) proposed auxiliary-PINNs for solving forward and inverse problems of integro-differential equations. These works highlight how recent research focuses on tailoring PINNs to solve specific problems by developing innovative models and methods.
>
> Given this context, it is important to note that the field lacks a single state-of-the-art PINN model that can be universally applied to all inverse problems. Instead, researchers evaluate how well each proposed model addresses the specific problems it is designed for. Consequently, comparing every model across all problems remains a significant challenge.
>
> In line with this trend, our work focuses specifically on the neutral inclusion problem, introducing a model called CoCo-PINNs that incorporates complex analysis techniques into PINNs.
> This model is designed to solve imperfect interface problems involving complex geometries and to design neutral inclusion, all without requiring datasets.
> Since CoCo-PINNs are tailored to this very specific problem, direct comparisons with other state-of-the-art PINN variants are challenging. Instead, we provide a detailed explanation of how classical PINNs encounter difficulties with this problem in terms of credibility, consistency, and stability, and demonstrate why our proposed method is more effective.
>
> We sincerely appreciate your feedback, as it allowed us to better clarify the scope, positioning, and focus of our proposed model. Thank you once again for raising this important point.
>
> ### W3. Some sections lack clarity
> We thank the reviewer for this helpful comment. We have revised the manuscript to address the points raised in the Questions part.
>
> ### Q1 (question). Explicit Formulation of the conformal map
> We thank the reviewer for this helpful comment. We improved the manuscript accordingly in lines 198 - 199.
>
> Formula (2) is valid for any simply connected bounded domain in two dimensions. One can obtain the conformal mapping coefficients $\gamma$ and $a_n$ for a given closed curve by numerical computation, as the references provided in line 199.
>
> ### Q2. Clarification of Known and Unknown Terms in Equation 1
> We thank the reviewer for this helpful comment.
> The inclusion $\Omega$, along with the conductivities $\sigma_c$ and $\sigma_m$, is given. The interface function $p$ and the direct problem solution $u$ are unknown and depend on the background solution $H$ and the inclusion $\Omega$. In particular, $u$ is determined by the interface function $p$ when $H$ and $\Omega$ are fixed.
> We added this information more precisely in lines 187 – 189.
>
> ### Q3. The connection between Theorem 1 and the proposed method
> Theorem 1 outlines the series expansion formula for $u$, as you mentioned. This formula already incorporates the series expansion information for $p$. Lines 210 - 212 explain this, and Appendix C provides a more detailed relation between them.
>
>
> ### Q4. Rigorous definition of neutral inclusion
> We thank the reviewer for this helpful comment. To improve the clarity, we added the definition of neutral inclusion in Section 2.
>
> ### Q5. Necessary condition for all considered interfaces
> Our method is based on conformal mapping. All examples in Section 4 are formulated using their conformal maps. In particular, all the interfaces we consider are simply connected, bounded domains having analytic boundaries (refer to lines 846 - 847 for the definition of analytic boundary).
> As stated in our response to Question 1, it is possible to numerically compute the conformal map for a given inclusion.
>
> ### Q6. Necessary condition for the background solution
> We thank the reviewer for this helpful comment.
> We specified the applied field $H$ in the definition of neutral inclusions (see lines 183 - 185).

---

> > ### Comment · Reviewer_JNhw · 2024-11-25
> >
> > Thank you for the clarifications. However, the second point remains unclear to me. If PINN has not been reported in the literature for addressing the neutral inclusion problem, why was it chosen as a baseline? I think it would be more appropriate to select a method that has been validated in the literature for this specific problem as the baseline.

---

> > > ### Author Response · Authors · 2024-11-26
> > > **Reply to additional remarks**
> > >
> > > Thank you for your thoughtful comment.
> > > We believe the validated methods you mentioned likely refer to numerical approaches such as the finite element method (FEM), boundary element method (BEM), and others. However, to the best of our knowledge, FEM has not been reported in the literature as a method for addressing imperfect interface problems in unbounded domains. Additionally, BEM requires handling highly singular integrals due to the mixed-type boundary conditions associated with imperfect interfaces, which presents significant challenges in numerically solving the direct problem.
> > > For these reasons, there are no other established baselines available for this problem besides classical physics-informed neural networks (PINNs). Consequently, we considered classical PINNs as a natural starting point for addressing the imperfect interface problem while simultaneously designing an interface function to make the neutral inclusion effect.
> > > This approach was inspired by several studies that reported successful results in solving forward and inverse problems, or purely inverse problems, using classical PINNs (e.g., Chen et al., 2020, Haghighata et al 2021, Jagtap et al 2022,...).
> > > Based on these considerations, we established classical PINNs as the baseline method for tackling the neutral inclusion problem. By observing the limitations and challenges of this approach, we developed CoCoPINN as an improved solution to overcome these difficulties.

---

> > > > ### Comment · Reviewer_JNhw · 2024-11-27
> > > >
> > > > Thanks for your reply. I would like to clarify that, in my opinion, the method validated in the literature for this specific problem is superior. As you mentioned, neither FEM nor PINN have been reported in the literature as a method for addressing this problem. Is this a new problem that hasn't been computed numerically before? If so, numerically computing this problem should be the biggest contribution of this paper, and there should be a new story. However, this point was not mentioned in the paper. Additionally, for unbounded domains, one can set a large domain of interest, as shown in Figure 2 of the paper, and apply proper boundary conditions to use FEM.

---

> > > > > ### Author Response · Authors · 2024-11-27
> > > > > **Reply to additional comments 2**
> > > > >
> > > > > We sincerely appreciate your thoughtful and detailed feedback on our paper.
> > > > > We do not claim that this is the first time a numerical solver has been proposed for solving inclusion problems with imperfect conditions.
> > > > > Regarding the direct problem of solving the interface problem, the analytical method proposed by Choi & Lim (2024) allows for a solution through matrix computations. We agree with your comment that FEM when appropriately adapted for unbounded domains, can address the direct problem of the imperfect interface problem.
> > > > > We emphasize that we focus on addressing the forward-inverse problem addressing the forward-inverse problem, which involves solving the PDEs while simultaneously identifying a specialized interface problem. We initially anticipated that classical PINNs could offer a suitable solution for the forward-inverse problem.
> > > > > However, our findings revealed that classical PINNs struggled to solve the forward problem as the complexity of the domain's shape increased. This motivated us to develop improvements that demonstrated superior performance, particularly in solving the inverse problem.

---

> > > > > > ### Author Response · Authors · 2024-12-04
> > > > > >
> > > > > > We hope our response has adequately and thoroughly addressed your concerns.
> > > > > >
> > > > > > If you find our clarifications satisfactory, we would be sincerely grateful if you could kindly reconsider your score.

---

### Official Review · Reviewer_nVyJ · 2024-10-31

**Soundness:** 2
**Presentation:** 3
**Contribution:** 3
**Rating:** 5
**Confidence:** 3

**Summary:**

The authors developed a Conformal Mapping Coordinates Physics-Informed Neural Networks (CoCo-PINNs) to solve the neutral inclusion problem. The theoretical analysis is thorough.

**Strengths:**

The theoretical analysis is thorough on the PINN aspect.

**Weaknesses:**

The results on the inverse part are not very impressive.

**Questions:**

1. In the neutral inclusion problem, the primary goal of this inverse problem is to determine the inclusion's geometry. However, the geometry identification results of Classical PINNs and CoCo-PINNs are quite similar, which is insufficient to demonstrate the proposed method's effectiveness in addressing the inverse problem.
2. What is the overhead of Classical PINNs and CoCo-PINNs for this problem.
3. What are the inclusion geometries for training.
4. Can the proposed method be applied for more complex inclusion geometries.

---

> ### Author Response · Authors · 2024-11-20
> **Response to Reviewer nVyJ**
>
> We thank the reviewer for the valuable comment. We provide some clarification as follows.
>
> ### Q1 (question). Effectiveness in geometry identification
> We want to clarify that our goal is not to determine the geometry of the inclusion but rather to design an interface function that ensures the inclusion behaves as a neutral inclusion, while simultaneously solving the forward problem simultaneously. Consequently, the inclusion geometry is assumed to be given, and our focus is on determining the interface parameter on the inclusion boundary.
>
> As future research, it would indeed be meaningful to generalize the proposed approach to a free boundary problem, where both the inclusion’s shape and the interface parameter are simultaneously determined to achieve the neutral inclusion effect.
>
> ### Q2. What is the overhead of Classical PINNs and CoCo-PINNs for this problem
> Both methods require similar computation times and are designed to simultaneously address the forward and inverse problems. However, our approach demonstrates higher performance compared to classical PINNs in the following aspects:
> Credibility assesses how closely the forward solutions from classical PINNs and CoCo-PINNs align with the analytical solution and quantifies any discrepancies.
> Consistency evaluates whether the interface functions produced by both methods exhibit a stable trend, consistently delivering similar outputs.
> Stability measures the robustness of both models under changes in conductivity variables.
>
> For instance, as shown in Figure 6, the forward solution of classical PINNs does not match the analytical solution, while the results of CoCo-PINNs do.
>
> ### Q3. What are the inclusion geometries for training?
> As mentioned in the Introduction lines 146–148, we do not require training data. Once the inclusion is provided, our PINNs simultaneously provide the interface function and the corresponding forward solution. After determining a specific interface function, it can be applied to other linear background solutions.
>
> ### Q4. Can the proposed method be applied to more complex inclusion geometries?
> The proposed approach is applicable to a simply connected bounded domain with an analytic boundary. However, the existence and stability of the inverse problem of determining an imperfect parameter that achieves neutral inclusion have not yet been theoretically verified. Furthermore, the performance may vary depending on the geometry of the inclusions, requiring further investigation.

---

> ### Author Response · Authors · 2024-11-26
> **Remind for a review.**
>
> Dear Reviewer nVyJ,
>
> We kindly remind you that we have submitted our response to your review comments. With the discussion period ending in few days, we would greatly appreciate it if you could let us know whether our response has addressed your concerns or if you have any additional feedback. Thank you again for your valuable comments and suggestions, which have been instrumental in improving our work.

---

> > ### Author Response · Authors · 2024-12-04
> >
> > We hope our response has adequately and thoroughly addressed your concerns.
> >
> > If you find our clarifications satisfactory, we would be sincerely grateful if you could kindly reconsider your score.

---

### Official Review · Reviewer_wjNp · 2024-11-01

**Soundness:** 3
**Presentation:** 2
**Contribution:** 2
**Rating:** 3
**Confidence:** 3

**Summary:**

This paper aims to improve how physics-informed neural networks (PINNs) solve the problem of modeling neutral inclusions, with the goal of inverse design. An "inclusion" is a deviation from the background medium in a field-solution problem, e.g., the heat equation.  Such inclusions result in perturbations of the surrounding field, which may be undesirable.  When a linear field is applied but the inclusion does not perturb the field, this is called a "neutral" inclusion.  This desirable property can sometimes be achieved by designing a coating for the interface bewteen the inclusion and the surround.  The goal of this paper is to modify PINNs so they can better model this kind of behavior and then also use them for inverse design of the interface.

The main idea is to use a conformal map on the exterior of the inclusion and thereby warp it in a way to more easily apply PINNs to complicated inclusion geometries; the map being conformal is desirable for reasoning about neutrality under linear applied fields.  The conformal map itself is fit a priori based on the inclusion geometry using methods from another paper.  A secondary point of the paper is using a Fourier basis rather than a neural net to represent the interface function.

The experiments investigated the quality and consistency of the solution on neutral inclusion problems for which an analytic solution is available.

**Strengths:**

The neutral inclusion problem is interesting.  Using a conformal map to warp space to make it easier to handle complicated geometries is a good (and well studied) idea.  Doing this for PINNs is reasonable.

**Weaknesses:**

The paper is not structured well.  It took me a long time to figure out what the paper was doing.  It did not do a good job explaining what neutral inclusions are to someone who is not already familiar with the concept.  Figure 1 was not illuminating in this regard.  Overall, this left me with some substantial uncertainty about whether I understood it.

I was a little disappointed when I realized that the conformal map is not part of the machine learning setup.  That's not to say that what the paper did is the wrong way to solve the problem, but the message of the paper ends up being fairly modest: "Do Choi & Lim (24) before using a PINN."

Conformal mappings to simplify solutions to PDEs is a well studied idea.  I don't know much about neutral inclusions specifically, but even for that problem a quick search reveals papers that use conformal maps, just not for PINNs.

The experiments are all on problems for which an analytic solution is available, which makes it difficult to motivate using a neural network.  I appreciate that this is valuable as one aspect of understanding the performance of the method, but the paper would be much stronger if it solved problems that demand numeric solutions and then, very importantly, compared to conventional numerical approaches.  A recent review paper has highlighted poor motivation and baselines in papers on PINNs:

McGreivy, Nick, and Ammar Hakim. "Weak baselines and reporting biases lead to overoptimism in machine learning for fluid-related partial differential equations." Nature Machine Intelligence (2024): 1-14.

In other words: it is not sufficient to compare your method to other PINNs. In the case of inverse problems, it's natural to ask for comparisons to adjoint approaches.

**Questions:**

How does this perform on problems that require numerical solutions?

What is the computational cost of the method relative to conventional numerical methods?

Out of curiousity: are spatially-varying coatings of the form studied here physically plausible?  Would they usually be done by, e.g., varying the thickness of a homogenous coating material?

**Details Of Ethics Concerns:**

I feel obligated to point out that a major use of the method in this paper is to design stealth/cloaking technology for weapons.  That is not the only use, but it is a significant application.

---

> ### Author Response · Authors · 2024-11-20
> **Response to Reviewer wjNp(1/2)**
>
> We thank the constructive comments from the reviewer. We provide some clarification as follows.
>
> ### W1(comment in weakness). Explaining what neutral inclusions are
> We thank the reviewer for this helpful comment. To improve the understanding for readers who are not already familiar with neutral inclusions, we have revised the introduction for greater clarity. In particular, we added Figure 1 in the introduction and the definition of neutral inclusions in Section 2.
>
> ### W2. Conformal map in machine learning setup
> The main goal of our paper is to identify an interface function that makes the inclusion a neutral inclusion, with the inclusion being given by a fixed conformal map. The machine learning setup incorporates this conformal mapping in selecting the collocation points and in expressing the imperfect parameter as a Fourier series. Conformal maps locally preserve angles, and it is helpful to deal with imperfect boundary conditions on the inclusion boundary.
>
> We did not directly adopt the methodology presented in Choi \& Lim’s paper to design PINNs. Instead, our design was inspired by their ideas and underlying philosophy, which suggest that inverse parameters can be more effectively determined by considering the coefficients of their series expansions, rather than using a direct approach with PINNs.
> Choi \& Lim’s results provide an analytic solution expressed through the series expansion coefficients of the interface function. We use this solely to test the credibility, not as part of the PINN design process.
>
> If you consider the problem involves finding both the appropriate shape of the inclusion and the interface function simultaneously, this represents a different perspective from what we focus on since in our study, we fix the inclusion $\Omega$ and find the interface function.
>
> ### W3. Conformal mapping for inclusion problems
> As you mentioned, the use of conformal mapping to simplify solutions to PDEs is well-studied, and it has also been employed to address inclusion problems, including neutral inclusions. We have added relevant references to the introduction in lines 106 - 108 and 112 - 113.
> Notably, in this paper, we extend this approach to the domain of machine learning by proposing a novel method that combines conformal maps with PINNs to solve the neutral inclusion problems.
>
> ### W3-4. Motivation using a neural network and Comparing with numerical approaches
> We thank the reviewer for this helpful comment. We modified lines 126 - 132 in the introduction to better explain the usage of the analytic solution results, where we highlight that the analytic direct solutions are only used to test the performance of the proposed method in finding the forward solution.
>
> Asymptotic approaches, such as in Kang\&Li (2019) and Choi\&Lim (2024), deal with coefficients of the asymptotic expansion for the direct solution. Consequently, these methods require analytical expressions for the direct solution. In this paper, by using a neural network, we focus on solution values rather than asymptotic approximations, as stated in lines 114 - 115 and 116 - 119. **The proposed method bypasses the need for an analytical expansion formula by designing the loss function including the values of the direct solution.**
>
> The use of conformal mapping to address inclusion problems in PDEs is a well-established approach with numerous analytical results. Developing machine learning methods that leverage conformal mapping and integrate these analytical results could open new avenues for studying inclusion problems and, more broadly, for advancing the theory of composite materials.
>
> As you mentioned, it would indeed be much more helpful to compare our approach to conventional numerical methods, such as finite element methods(FEM), boundary elementary methods(BEM), and others.
> However, to the best of our knowledge, FEM has not been reported in the literature for addressing imperfect interface problems in unbounded domains. Moreover, BEM involves handling highly singular integrals due to the mixed-type boundary conditions associated with imperfect interfaces, which pose significant challenges in numerically solving the direct problem with BEM. For these reasons, we used the analytical solution to evaluate the performance of our proposed method for solving the direct problem.
>
> Finally, our approach is designed to simultaneously address the forward and inverse problems. Hence, we provided the comparison with the classical PINNs, which provide the forward-inverse solutions. This is the main motivation behind our use of neural networks, and we sincerely thank you once again for highlighting the key aspects of our proposed method.

---

> > ### Comment · Reviewer_wjNp · 2024-11-25
> >
> > I confirm that I have read this response.

---

> > > ### Author Response · Authors · 2024-11-26
> > >
> > > Thank you for confirming that you have read our response. We hope it has addressed your concerns and would greatly appreciate it if you could kindly let us know and adjust the score if appropriate.

---

> > > > ### Author Response · Authors · 2024-12-04
> > > >
> > > > We hope our response has adequately and thoroughly addressed your concerns.
> > > >
> > > > If you find our clarifications satisfactory, we would be sincerely grateful if you could kindly reconsider your score.

---

> ### Author Response · Authors · 2024-11-20
> **Response to Reviewer wjNp(2/2)**
>
> ### Q1 (question). How does this perform on problems that require numerical solutions?
> As discussed, classical methods like FEM and BEM face significant challenges in addressing imperfect interface problems. Our method, however, offers an effective alternative.
> We also anticipate that it will be particularly valuable in cases where no analytic or conventional numerical solution is readily available.
>
> ### Q2. What is the computational cost of the method relative to conventional numerical methods?
> We thank the reviewer for the valuable comment.
> Currently, no numerical methods have been presented to simultaneously handle both the forward and inverse problems for this imperfect interface problem.
> In the future, if researchers develop neutral inclusion problems using FEM or BEM, they can compare our method. For reference, both CoCo-PINNs and classical PINNs use the same model size of 2-20-20-20-20-1 for the forward solution, achieving results in about 20 minutes. However, for the interface function, classical PINNs require a model size of 2-20-20-20-20-2, while CoCo-PINNs use 10 complex coefficients.
>
> ### Q3. Are spatially-varying coatings of the form studied here physically plausible?
> The imperfect boundary condition is a mathematical model for membrane structures, such as cell membranes, which consists of a thin layer between two materials, such as cell membranes. In this article, the imperfect parameter addresses the LC-type interface, where the imperfect boundary condition is determined by the limit of $k_c/t$ as $k_c, t\rightarrow 0$. Here, $k_c$ and $t$ represent the variable conductivity and the constant thickness of the thin interface layer, respectively, and the resulting limit is allowed to vary along the interface. For further details, we refer to Benveniste\&Miloh, J. Mech. Phys. Solids (1999), cited in the reference list.
>
> ### Regarding Flag for ethics review
> Thank you for pointing out the aspects related to ethics that we had previously overlooked. In response, we added a new section on ethics, positioned just before the references section in the manuscript. Additionally, in lines 72–78 of the manuscript, we have clarified the motivation behind this study and provided more detailed explanations and references to better explain the various research directions in the study of neutral inclusions and invisibility cloaking.
>
> While some studies on neutral inclusions and invisibility cloaking that involves **wave propagation** might potentially be associated with the design of stealth or cloaking technologies for weapons, there are numerous other directions aimed at positive applications, including noise reduction, enhanced sensing, and improved communication systems. Specifically, our research focuses on the **quasi-static conductivity equation**, with relevant applications in designing reinforced or embedded composite materials that preserve the original stress field of the material without inclusions and avoid stress concentrations. We emphasize that this research is intended solely for constructive and ethical purposes.

---

### Author Response · Authors · 2024-11-20
**General Response**

# General Response

We would like to express our gratitude to all the reviewers for their careful reading of the manuscript and their invaluable feedback and suggestions, which have greatly improved our paper.

The insights provided by the reviewers were essential in revising our manuscript, and we are pleased to see that the reviewers appreciated several aspects of our proposed approach. Specifically, we are grateful for the acknowledgment of our theoretical analysis (Reviewers nVyJ, JNhw), the use of conformal mapping to address complex-shaped inclusions (Reviewer wjNp), and the recognition that the proposed method does not require training data to solve the neutral inclusion problem (Reviewer JNhw).

We are open to addressing any concerns or questions raised and welcome the opportunity to provide further clarification. All the suggestions raised by the reviewers have been incorporated into the revised version of the manuscript. The revised lines are highlighted in red in the revised manuscript.

It is currently over 10 pages long, but we plan to reduce it to under 10 pages in the camera-ready version.

In response to the reviewers' comments, we have made the following changes to the manuscript.

- We improved the introduction to include more relevant references and better explain the motivation for the proposed approach. Additionally, we emphasized that the analytic direct solution is used to test the performance of the direct solution.
- We added the definition of neutral inclusions (see **Definition 1** in Section 2) and **Figure 1** to provide a clearer explanation of the concept. We also highlighted the motivation for studying neutral inclusions and their applications in the introduction.
- Minor corrections were made to address misprints, which are not marked in color in the manuscript.
- An ethics statement was added.

For more detailed explanations, please refer to our individual responses to each reviewer. If there are any concerns, uncertainties, or questions from the reviewers, we are happy to address them and provide further elaboration during the discussion phase. We look forward to communicating with you.

---

### Meta-Review · Area_Chair_psfq · 2024-12-15

**Metareview:**

The paper proposes using conformal maps to aid PINNs to solve the neural inclusion problem. The approach arguably simplifies the task for PINNs, by avoiding the complex geometries associated with inclusion. The reviewers mostly agree that the proposed approach makes sense and is quite reasonable.

At a high level, two aspects led to concerns regarding the work: (1) Novelty of the approach: using conformal maps has been used in the relevant literature and PINNs are widely studied; the current work effectively lines up these two ideas. (2) Evaluation: There was extensive discussions on the simplicity of the experiments and whether the proposed approaches will indeed be effective in realistic problems. The authors acknowledge the concerns, and gave point-by-point responses to the reviewer concerns. The concerns unfortunately did not get satisfactorily resolved.

**Additional Comments On Reviewer Discussion:**

One of the reviewers (NvYj) did not engage in discussions. The others acknowledged the response, and one reviewer engaged with the authors. No points were changed.

---

### Decision · Program_Chairs · 2025-01-22

Reject